*Method*

# An immunohistochemical atlas of necroptotic pathway expression

Shene Chiou[1,2,8], Aysha H Al-Ani[1,2,3,8], Yi Pan [1], Komal M Patel[1], Isabella Y Kong[4],
Lachlan W Whitehead[1,2], Amanda Light[1], Samuel N Young[1], Marilou Barrios[1,2], Callum Sargeant[1,2],
Pradeep Rajasekhar [1,2], Leah Zhu [1], Anne Hempel[1], Ann Lin[1], James A Rickard[1,5], Cathrine Hall[1],
Pradnya Gangatirkar[1], Raymond KH Yip [1,2], Wayne Cawthorne [1,2], Annette V Jacobsen [1,2],
Christopher R Horne[1,2], Katherine R Martin [1,2], Lisa J Ioannidis [1,2], Diana S Hansen [1,2,6],
Jessica Day[1,2,3], Ian P Wicks[1,2], Charity Law[1,2], Matthew E Ritchie[1,2], Rory Bowden[1,2],
Joanne M Hildebrand [1,2], Lorraine A O'Reilly [1,2], John Silke [1,2], Lisa Giulino-Roth[4], Ellen Tsui [1],
Kelly L Rogers[1,2], Edwin D Hawkins[1,2], Britt Christensen[1,2,3], James M Murphy [1,2,7,9]✉ &
André L Samson [1,2,9]✉

## Abstract

**Necroptosis is a lytic form of regulated cell death reported to contribute to inflammatory diseases of the gut, skin and lung, as well as ischemic-reperfusion injuries of the kidney, heart and brain. However, precise identification of the cells and tissues that undergo necroptotic cell death in vivo has proven challenging in the absence of robust protocols for immunohistochemical detection. Here, we provide automated immunohistochemistry protocols to detect core necroptosis regulators – Caspase-8, RIPK1, RIPK3 and MLKL – in formalin-fixed mouse and human tissues. We observed surprising heterogeneity in protein expression within tissues, whereby short-lived immune barrier cells were replete with necroptotic effectors, whereas long-lived cells lacked RIPK3 or MLKL expression. Local changes in the expression of necroptotic effectors occurred in response to insults such as inflammation, dysbiosis or immune challenge, consistent with necroptosis being dysregulated in disease contexts. These methods will facilitate the precise localisation and evaluation of necroptotic signaling in vivo.**

**Keywords** IBD; Necroptosis; Immunohistochemistry; RIPK3; MLKL
**Subject Categories** Autophagy & Cell Death; Methods & Resources

## Introduction

The necroptotic cell death pathway leads to cell lysis and expulsion of cellular contents into the extracellular milieu, which in turn provokes an innate immune response. Necroptosis is considered to be an altruistic cell death pathway whose principal role is to protect the host from pathogens (Fletcher-Etherington et al, 2020; Liu et al, 2021; Palmer et al, 2021; Pearson et al, 2017; Petrie et al, 2019; Upton et al, 2010; Yeap and Chen, 2022; Zhang et al, 2020). Despite this, it is the aberrant functions of necroptosis associated with inflammatory diseases that have spurred interest in its underlying mechanisms and therapeutic prospects (Choi et al, 2019; Fang et al, 2021). Studies of mice lacking the terminal effectors of the pathway – RIPK3 (Receptor-interacting protein kinase-3) or MLKL (Mixed-lineage kinase domain-like) – have led to the concept that excess necroptosis drives a range of inflammatory pathologies in organs including the skin, gut, brain, heart, lung, kidney and testes (Devos et al, 2020; Gunther et al, 2011; Ito et al, 2016; Li et al, 2017; Linkermann et al, 2013; Lu et al, 2021; Luedde et al, 2014; Naito et al, 2020). However, many of these attributions have been disputed (Dominguez et al, 2021; Newton et al, 2016; Wang et al, 2020b), likely reflecting an evolving understanding of the pathway and the limited availability of validated reagents to interrogate necroptosis in pathological specimens.

The core signaling axis of the necroptotic pathway has been well-defined and can be activated in response to a variety of inflammatory cues including ligation of death, Toll-like or pathogen-pattern receptors (Chen et al, 2022b; Cho et al, 2009; Degterev et al, 2008; He et al, 2011; He et al, 2009; Kaiser et al, 2013; Kaiser et al, 2011). Caspase-8 is a critical negative regulator of

[1]Walter and Eliza Hall Institute of Medical Research, Parkville, Australia. [2]University of Melbourne, Parkville, Australia. [3]Royal Melbourne Hospital, Parkville, Australia. [4]Pediatric Hematology/Oncology, Weill Cornell Medical College, New York, USA. [5]Austin Hospital, Heidelberg, Australia. [6]Monash Biomedicine Discovery Institute, Department of Microbiology, Monash University, Clayton, Australia. [7]Drug Discovery Biology, Monash Institute of Pharmaceutical Sciences, Monash University, Parkville, Australia. [8]These authors contributed equally: Shene Chiou, Aysha H Al-Ani. [9]These authors jointly supervised this work: James M Murphy, André L Samson. ✉E-mail: jamesm@wehi.edu.au; samson.a@wehi.edu.au

necroptotic signaling (Kaiser et al, 2011), whereby its deletion or loss-of-function promotes oligomerization of RIPK1 (Receptor-interactor protein kinase-1), TRIF (TIR domain-containing adapter molecule 1) and/or ZBP1 (Z-DNA-binding protein 1) (He et al, 2011; Kaiser et al, 2013; Newton et al, 2016). This oligomeric structure, otherwise known as the necrosome, promotes activation of the downstream effectors RIPK3 and MLKL (Samson et al, 2021b). RIPK3 recruits the MLKL pseudokinase to the necrosome, where it phosphorylates MLKL to provoke a conformational change, release from the necrosome, oligomerization and trafficking to the plasma membrane (Garnish et al, 2021; Murphy et al, 2013; Samson et al, 2020; Sun et al, 2012; Wang et al, 2014; Zhao et al, 2012). At the plasma membrane, accumulation of activated MLKL to a critical threshold level is required for membrane permeabilization via a poorly understood mechanism that brings about the cell's demise (Chen et al, 2014; Hildebrand et al, 2014; Samson et al, 2020).

As our understanding of the necroptosis pathway has grown, new tools and protocols have been developed to study necroptotic signaling in fixed cultured cells (Rodriguez et al, 2016; Samson et al, 2021a; Wang et al, 2014; Webster et al, 2018). However, robust procedures for assessing the necroptotic pathway in tissues are still lacking, often leading to contradictory reports in the literature and misattributions of necroptotic pathologies. Here, we report automated immunostaining protocols for detecting Caspase-8, RIPK1, RIPK3, and MLKL in mouse formalin-fixed paraffin-embedded tissues. These procedures have enabled the assembly of an atlas of necroptotic pathway expression in mouse tissues under basal conditions and during innate immune challenge. While the necroptosis machinery is rarely expressed in cell types other than short-lived barrier cells, sterile inflammation increased RIPK3 expression in the gut and liver, broadly predisposing multiple cell types to necroptotic death. In contrast, the elimination of the intestinal microflora diminished the expression of RIPK3 and MLKL to reduce necroptotic propensity in the gut. RIPK3 is also uniquely upregulated in splenic germinal centers suggesting it may have a non-necroptotic role in humoral immunity. Furthermore, we present robust protocols for detecting human Caspase-8, RIPK1, RIPK3, and MLKL and illustrate their utility for detecting dysregulated necroptosis in biopsies from patients with inflammatory bowel disease (IBD). Collectively, these protocols will empower the definitive evaluation of where and when necroptosis occurs in vivo in health and disease.

# Results

## Standardized immunohistochemical detection of the necroptotic pathway in mouse tissues

We recently compiled a toolbox of immunofluorescence assays to detect necroptotic signaling in cells (Samson et al, 2021a). This toolbox requires the use of: (1) non-crosslinking fixatives and (2) gene knockouts to account for non-specific signals; requirements that often cannot be met when immunostaining tissues. Here we aimed to develop robust immunohistochemistry protocols to detect the necroptotic pathway in formalin-fixed paraffin-embedded mouse tissues. Embedding and immunostaining was performed in an automated manner (see Methods and Protocols) to allow reliable

and scalable detection of the necroptotic pathway, and to lessen the future need to account for non-specific immunosignals using appropriate gene knockout controls. The specificity of thirteen monoclonal antibodies against Caspase-8, RIPK1, RIPK3, or MLKL was first tested by immunoblotting spleen homogenates from wild-type versus knockout mice (Appendix Fig. S1). The intensity and specificity of antibodies for immunohistochemistry was then iteratively optimized across 21 conditions (see Methods and Protocols and Appendix Fig. S1). At each optimization step, immunohistochemical signals from the spleen of wild-type versus knockout mice were quantified (Fig. 1Ai), ratioed (Fig. 1Aii), and integrated to yield an index of performance (Fig. 1Aiii). For example, this pipeline improved the detection of RIPK1 with the monoclonal antibody D94C12 by approximately three orders of magnitude (Appendix Fig. S1). In total, seven automated immunohistochemistry protocols to detect mouse Caspase-8, RIPK1, RIPK3, or MLKL were developed (Fig. 1B). The detection of Caspase-8, RIPK1, and RIPK3 using these immunohistochemistry protocols (Fig. 1B) closely aligned with the abundance of these proteins across multiple tissues as measured by high-resolution quantitative mass spectrometry (Fig. 1C), indicating both specificity and sensitivity. Despite many rounds of optimization with three specific anti-MLKL antibodies, mouse MLKL remained difficult to detect via immunohistochemistry in all tissues except the spleen (Fig. 1B,D; Appendix Fig. S1).

## Basal expression of the necroptotic pathway is restricted to fast-cycling immune barriers

The immunohistochemical profile of Caspase-8, RIPK1, RIPK3, and MLKL across seven different organs suggested that expression of the necroptotic pathway is heavily restricted in unchallenged mice (Fig. 1D). For example, RIPK3+ cells were scarce in the kidney and heart, and RIPK3 was undetectable in the brain (Fig. 1D). By comparison, co-expression of Caspase-8, RIPK1, and RIPK3 was evident in intestinal epithelial cells, some splenic regions and Kupffer cells (Fig. 2A). The *Tabula Muris* single-cell RNA sequencing dataset supports the conclusion that expression of the necroptotic pathway is highly restricted in mice (Fig. EV1A; (Tabula Muris et al, 2018)). Transcript expression of the necroptotic effectors, MLKL and RIPK3, was below detection limits in kidney epithelial, cardiac muscle, and resident brain cells, but was frequently detected in progenitor and immune barrier cell populations (Fig. EV1A; (Tabula Muris et al, 2018)).

Close inspection of sites where the necroptotic pathway was constitutively expressed showed unexpected layers of spatial regulation (Fig. 2A,B). In the epithelial barrier of the ileum, Caspase-8 expression was lower in crypts and higher at villi tips, whereas RIPK3 levels peaked in the transit-amplifying region and decreased towards the villus tip (Fig. 2Ai,Bi). In the colonic epithelial barrier, both Caspase-8 and RIPK3 exhibited higher expression in the base of the crypt and decreased towards the tip of the crypt (Fig. 2Aii,Bii). It is noteworthy that expression patterns of Caspase-8 and RIPK3 differ between the small and large intestine because these organs exhibit distinct cell death responses to the same inflammatory stimuli (e.g. TNF) or genetic deficiency (e.g., deletion of *Casp8* or *Fadd*) (Bader et al, 2023; Schwarzer et al, 2020; Tisch et al, 2022; Zelic et al, 2018). In the liver, Kupffer cells expressed Caspase-8, RIPK1, and RIPK3 (arrowheads

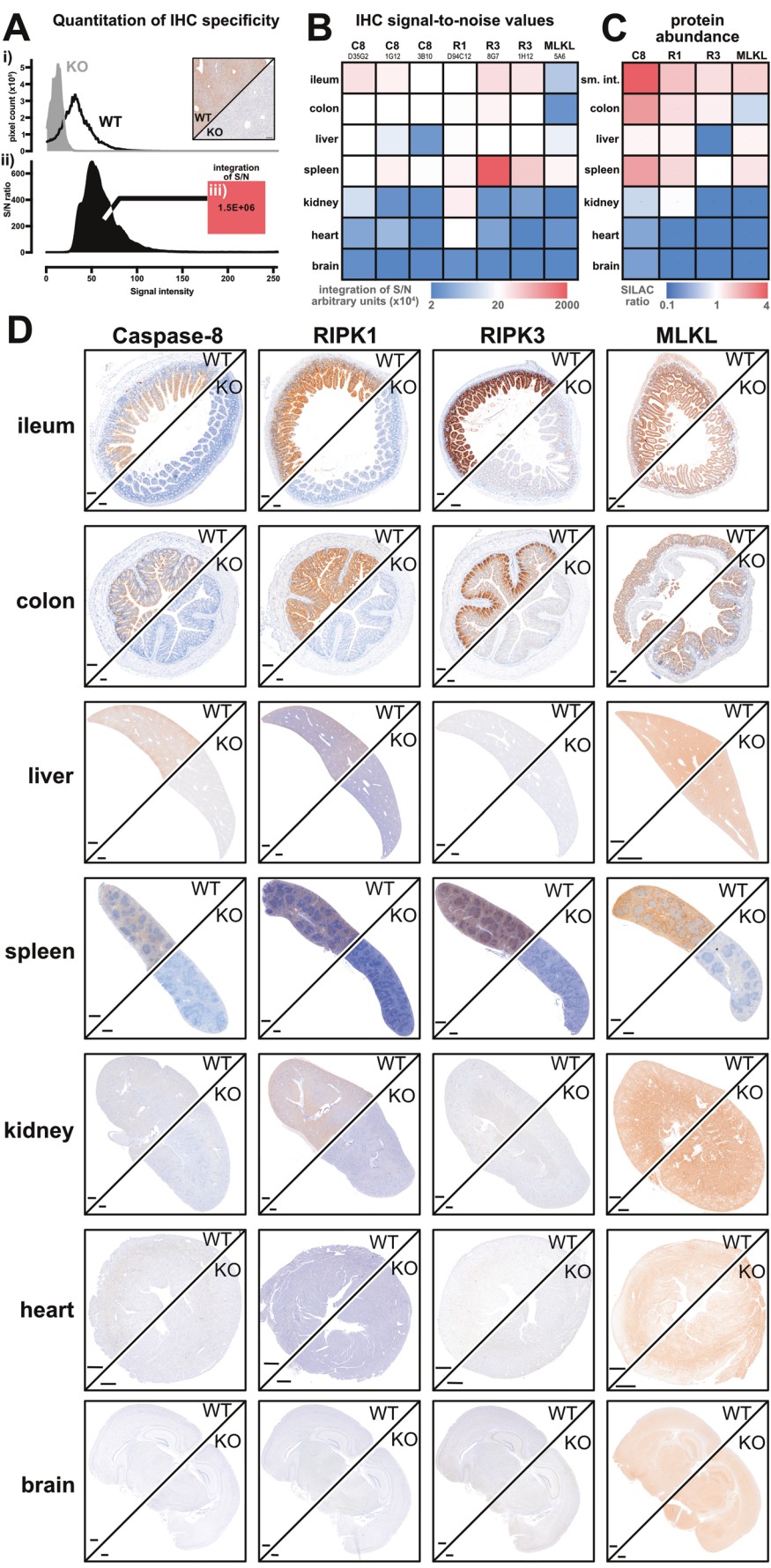

◄

**Figure 1. Automated immunohistochemistry shows constitutive necroptotic pathway expression is restricted.**

(A) To gauge immunohistochemistry performance, immunosignals from wild-type (WT) versus knockout (KO) tissue were deconvoluted, (i) pixel intensities plotted, (ii) ratioed to yield a signal-to-noise (S/N) histogram, and then (iii) integrated. (B) Heatmap shows relative integrated S/N values from seven automated immunohistochemistry protocols across seven tissues. Column headers indicate the antibody target clone name. Data were representative of $n \geq 3$ for each target and tissue. (C) Heatmap depicts relative protein abundance values as SILAC (stable isotope labeling by amino acids in cell culture) ratios measured by (Geiger et al, 2013); the lowest value was assigned as 0.1 because 0 is below the detection limit. (D) Immunosignals of Caspase-8, RIPK1, RIPK3, and MLKL in wild-type versus the appropriate knockout (KO) tissue from $Mlkl^{-/-}$ or $Casp8^{-/-}Ripk3^{-/-}$ or $Casp8^{-/-}Ripk1^{-/-}Ripk3^{-/-}$. Data were representative of $n \geq 3$ for each target and tissue. Scale bars are 500 μm. Related to Appendix Fig. S1. Source data are available online for this figure.

Fig. 2Aiii,Biii), whereas hepatocytes expressed Caspase-8 and RIPK1, but not RIPK3. Interestingly, Caspase-8 levels were higher in pericentral hepatocytes than in peri-portal hepatocytes (arrow Fig. 2Aiii,Biii). Zonation of the necroptotic pathway was also evident in the spleen, with Caspase-8, RIPK1, RIPK3, and MLKL levels peaking in the marginal zone, where circulating antigens are trapped for immune presentation (Fig. 2Aiv,EV1B). Prior spatial transcriptomics data support the conclusion that necroptotic potential is zoned along the intestinal crypt-to-villus axis (Fig. EV1C; (Moor et al, 2018)) and along the hepatic central-to-portal axis (Fig. EV1D; (Ben-Moshe et al, 2019)). We also used spatial transcriptomics on mouse spleen to confirm that necroptotic potential peaks in the marginal zone (Fig. EV1E–G).

The expression of RIPK3 appears to be under particularly strict spatial control. For example, in the ileum, RIPK3 levels were high in fast-cycling epithelial progenitors, but low in adjacent, terminally differentiated Paneth cells (open arrowhead Fig. 2Ai). Published single-cell transcriptomics data supports the conclusion that Paneth cells express low levels of RIPK3 under basal conditions (Haber et al, 2017). As another example of differential expression, RIPK3 levels were high in fast-cycling colonic epithelial cells, but undetectable in slow-cycling renal epithelial cells (Fig. 1D). These observations suggest that RIPK3 expression is linked to cell turnover. Indeed, across 103 cell ontologies in the *Tabula Muris* dataset, gene expression of cell cycle markers *Top2a* and *Mki67* correlated with the expression of *Ripk3*, but not *Ripk1* (Figs. 2C and EV1A; (Tabula Muris et al, 2018)). Prior cell culture studies further suggest that the expression and function of RIPK3 fluctuates during the mitotic cell cycle (Gupta and Liu, 2021; Liccardi et al, 2019).

Altogether, by applying a set of optimized immunohistochemistry protocols to multiple organs, we have found that the necroptotic pathway is preferentially expressed at fast-cycling immune barriers under basal conditions. Such targeted expression is consistent with the evolutionary origin of necroptosis being an anti-pathogen defense measure (Palmer et al, 2021; Petrie et al, 2019; Upton et al, 2010). We further find that necroptotic potential is spatially graded along barriers such as the intestinal mucosa. These gradations in the availability of cell death mediators along barriers likely allow different cell death programs to be flexibly deployed against invading pathogens (Cook et al, 2014; Doerflinger et al, 2020).

## Inflammation, dysbiosis, or immune challenge trigger local changes in RIPK3 expression

To demonstrate scalability, we used automated immunohistochemistry to characterize the expression of Caspase-8, RIPK1 and RIPK3 across six tissues during TNF-induced systemic inflammatory response syndrome (SIRS)—a widely-used model of RIPK-dependent pathology (Fig. 3A; (Duprez et al, 2011; Harris et al, 2017; Newton et al, 2016; Newton et al, 2014; Zelic et al, 2018)). Littermate wild-type mice were intravenously administered TNF or vehicle and tissues were harvested 9 h later when symptoms such as hypothermia were manifesting (Fig. 3B). No major changes to Caspase-8 or RIPK1 expression were observed after TNF administration, except for an unidentified population of RIPK1-expressing cells appearing at the onset of apoptosis in lymphoid tissues (Fig. 3C,D; arrowhead). By comparison, RIPK3 was upregulated in intestinal epithelial cells (Fig. 3E,F), certain vascular beds (Fig. 3G,H), and in the liver (Fig. 3I,J); the main sites where RIPK1- and RIPK3-mediated signaling during SIRS has been implicated by knock-in and knockout mouse studies (Duprez et al, 2011; Newton et al, 2016; Zelic et al, 2018). In contrast, RIPK3 levels were not increased in resident cells of the kidney or heart in TNF-treated mice. Our data, therefore, suggest that targeted upregulation of RIPK3 in resident cells of the gut and liver underlies RIPK-mediated pathology in SIRS. TNF-treatment also changed the pattern of RIPK3 expression in the intestine, potentially skewing cell death responses in the inflamed gut (Fig. 3E,F). It was surprising that RIPK3 was detected in peri-portal hepatocytes after TNF administration, given that RIPK3 is epigenetically silenced in hepatocytes under basal conditions (Preston et al, 2022). Collectively, our immunohistochemical characterization of the SIRS mouse model leads us to propose that RIPK3 is regulated akin to a positive acute phase reactant, with hepatic and intestinal expression that rapidly increases in response to inflammation. In support of this notion, we find that TNF-treatment increased the levels of RIPK3 in serum (Fig. 3K,L). Moreover, RIPK1-inhibition prevented both TNF-induced hypothermia and the release of RIPK3 into the blood (Fig. 3K,L). These data raise the exciting possibilities that RIPK3 is a novel acute phase reactant, and that circulating levels of RIPK3 are a surrogate measure of RIPK1-mediated signaling.

Next, we addressed whether microbiota-depletion affects necroptotic pathway expression. This question was prompted by studies showing that antibiotics offer protection in various models of intestinal necroptosis (Bader et al, 2023; Gunther et al, 2015; Li et al, 2020; Xie et al, 2020). As shown in Fig. 4A, a litter of wild-type mice was split into two cages and the water for one cage was supplemented with antibiotics for 6 days. As expected, the cecum of antibiotic-treated mice was enlarged and canonical anti-microbial factors such as lysozyme and REG-3β were reduced in the ileum, but not the spleen, of antibiotic-treated mice (Fig. 4B,C). These predictable responses to microbiota-depletion also coincided with a lowering of RIPK3 and MLKL gene and protein expression in the ileum, but not the spleen (Fig. 4B,C; Appendix Fig. S2). In

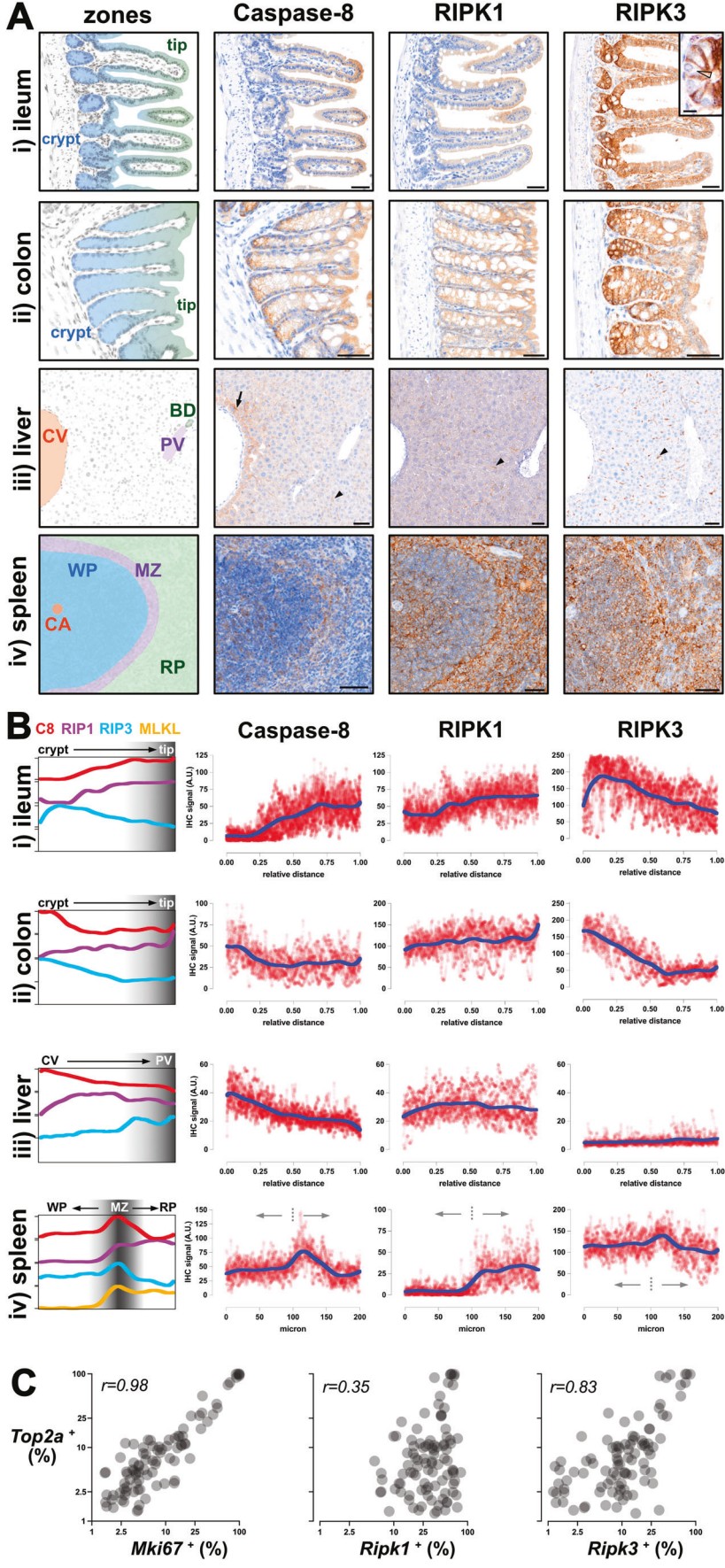

Figure 2. Necroptotic potential is spatially graded across tissue zones.

(A) Immunosignals of Caspase-8, RIPK1, and RIPK3 from wild-type mouse ileum (i), colon (ii), liver (iii), and spleen (iv). The crypt base (crypt), villi/crypt tip (tip), central vein (CV), portal vein (PV), bile duct (BD), central artery (CA), white pulp (WP), marginal zone (MZ), and red pulp (RP) are annotated. Inset of immunostaining in the ileum shows lower RIPK3 expression in Paneth cells (open arrowhead) relative to neighboring cells. Arrow shows pericentral hepatocytes that express higher levels of Caspase-8. Closed arrowheads show Caspase-8+ RIPK1+ RIPK3+ Kupffer cells. Scale bars are 50 μm, except for the 10 μm scale bar in the inset. Data were representative of $n \geq 3$ for each target and tissue. (B) Relative expression levels of Caspase-8, RIPK1, and RIPK3 (and splenic MLKL; Fig. EV1B) along the indicated tissue axes. Red datapoints indicate immunosignal intensities, and the overlaid dark blue line indicating the LOWESS best-fit along $N = 20$ axes per tissue. Best-fit curves are superimposed in the left-most column. The dashed line indicates the boundary between the splenic white pulp and the marginal zone. Data were representative of $n > 3$ mice per target per tissue. (C) Scatterplots where each dot represents a different cell ontology from the Tabula Muris dataset (Tabula Muris et al, 2018). The percent of cells within each ontology that expressed Mki67, Ripk1, or Ripk3 was plotted against that of Top2a. Pearson correlation coefficient values are shown. Related to Fig. EV1. Source data are available online for this figure.

contrast, Caspase-8 gene and protein expression in the ileum were unaffected by microbiota-depletion, whereas ileal RIPK1 protein levels were increased by antibiotic treatment (Fig. 4B,C; Appendix Fig. S2). Similar trends were observed by immunohistochemistry, with epithelial Caspase-8 expression remaining constant, while RIPK1 levels were elevated and RIPK3 expression reduced in the crypt and transit-amplifying regions of the ileum in antibiotic-treated mice (Fig. 4D). Unexpectedly, immunohistochemistry also showed that microbiota-depletion triggered cytoplasmic accumulations of RIPK1 and RIPK3 in enterocytes at villi tips (Fig. 4D; arrowheads). These RIPK1+ RIPK3+ Caspase-8- clusters are unlikely to be necrosomes, as no corresponding phospho-activation of RIPK1 or MLKL was observed (Appendix Fig. S2). Instead, these clusters may be due to a microbe-related function, such as lipopolysaccharide handling, that is preferentially performed by enterocytes at villi tips (Berkova et al, 2023; Ge et al, 2000). These changes to RIPK1/3 could also be related to the reduced epithelial turnover that accompanies microbiota-depletion (Park et al, 2016). Overall, we find that expression of the necroptotic pathway responds locally to changes in the microbiome. This response is spatially restricted to the small intestine, zoned along the crypt-to-villus axis, and warrants further investigation given that dysbiosis often occurs in cell death-associated gut disorders such as Crohn's disease (Gevers et al, 2014).

Lastly, we used automated immunohistochemistry to uncover a potential non-necroptotic role for RIPK3 in adaptive immunity. We immunized wild-type mice with the model ligand, NP-KLH (4-hydroxy-3-nitrophenylacetyl hapten conjugated to keyhole limpet hemocyanin), and harvested tissues 5 or 14 days later when antigen-specific antibody responses were detectable (Fig. EV2A). No changes in the levels or zonation of Caspase-8, RIPK1, or RIPK3 were noted in the ileum of immunized mice, and no marked differences in the expression of Caspase-8, RIPK1, or MLKL were observed in the spleen 14 days after immunization. However, RIPK3 levels were markedly elevated in Ki67+ germinal centers (arrowheads Fig. EV2B). This finding suggests that RIPK3 may have a non-necroptotic role in antibody production. To investigate this possibility, Ripk3+/+ and Ripk3−/− littermate mice were immunized with NP-KLH and humoral immune responses were measured in blood and spleen (Fig. EV2C–J). RIPK3-deficiency did not alter the circulating levels of antigen-specific antibodies 5 days after immunization (Fig. EV2C,D). Similarly, RIPK3-deficiency did not influence the number of class-switched B cells in the spleen (Fig. EV2E,F), the number of antigen-specific plasma cells in the spleen (Fig. EV2G), or the amount of circulating antigen-specific antibodies 14 days after immunization (Fig. EV2H–J). Thus, consistent with prior

studies (Newton et al, 2004), RIPK3 does not overtly affect early antigen-specific antibody responses. Future studies should explore the role of RIPK3 in splenic germinal centers, especially given that RIPK3 has a mechanistically undefined non-necroptotic role during lymphoproliferative disease (Alvarez-Diaz et al, 2016).

In summary, by employing a toolbox of automated immuno-histochemical stains, we find that expression of the necroptotic pathway, in particular RIPK3, is responsive to inflammation, dysbiosis, or immunization. These context-specific changes are tightly regulated across space and time, underscoring the need for robust, scalable, in situ assays to pinpoint necroptotic pathway expression and activation.

## Automated immunohistochemical detection of the human necroptotic pathway

Important differences exist between the human and mouse necroptotic pathways (Chen et al, 2013; Davies et al, 2020; Horne et al, 2023; Petrie et al, 2018; Samson et al, 2021b; Sun et al, 2012; Tanzer et al, 2016). For instance, the primary sequence of RIPK3 and MLKL are poorly conserved between species (Horne et al, 2023), and humans uniquely express Caspase-10 which likely negates necroptotic signaling (Ramirez and Salvesen, 2018; Tanzer et al, 2017). Thus, in parallel to developing assays for the murine necroptotic pathway, seventeen antibodies against Caspase-8, Caspase-10, RIPK1, RIPK3 or MLKL were tested on wild-type versus knockout HT29 human cells via immunoblot, and then iteratively optimized for immunohistochemistry on formalin-fixed paraffin-embedded pellets of these same cell lines (see Methods and Protocols and Appendix Fig. S3). While Caspase-10 remained refractory to immunohistochemical detection, six automated immunohistochemistry protocols were developed for human Caspase-8, RIPK1, RIPK3 or MLKL (Appendix Fig. S3).

These immunohistochemistry protocols detected diffuse cyto-plasmic signals for Caspase-8, RIPK1, RIPK3, and MLKL in cells under resting conditions (Fig. 5A–C), and in cells undergoing TNF-induced apoptosis (via co-treatment with TNF (T) and a Smac mimetic (S); Fig. 5). Conversely, immunohistochemistry detected intracellular clusters of Caspase-8, RIPK1, RIPK3, or MLKL in cells undergoing TNF-induced necroptosis (via co-treatment with T and S and the caspase inhibitor IDN-6556 (I); arrowhead Fig. 5A). These intracellular clusters are presumed to be necrosomes because they resemble prior images of necrosomes (Chen et al, 2022b; Samson et al, 2021a; Samson et al, 2020; Sun et al, 2012) and because orthogonal approaches show that Caspase-8, RIPK1, RIPK3, and MLKL are recruited to necrosomes during

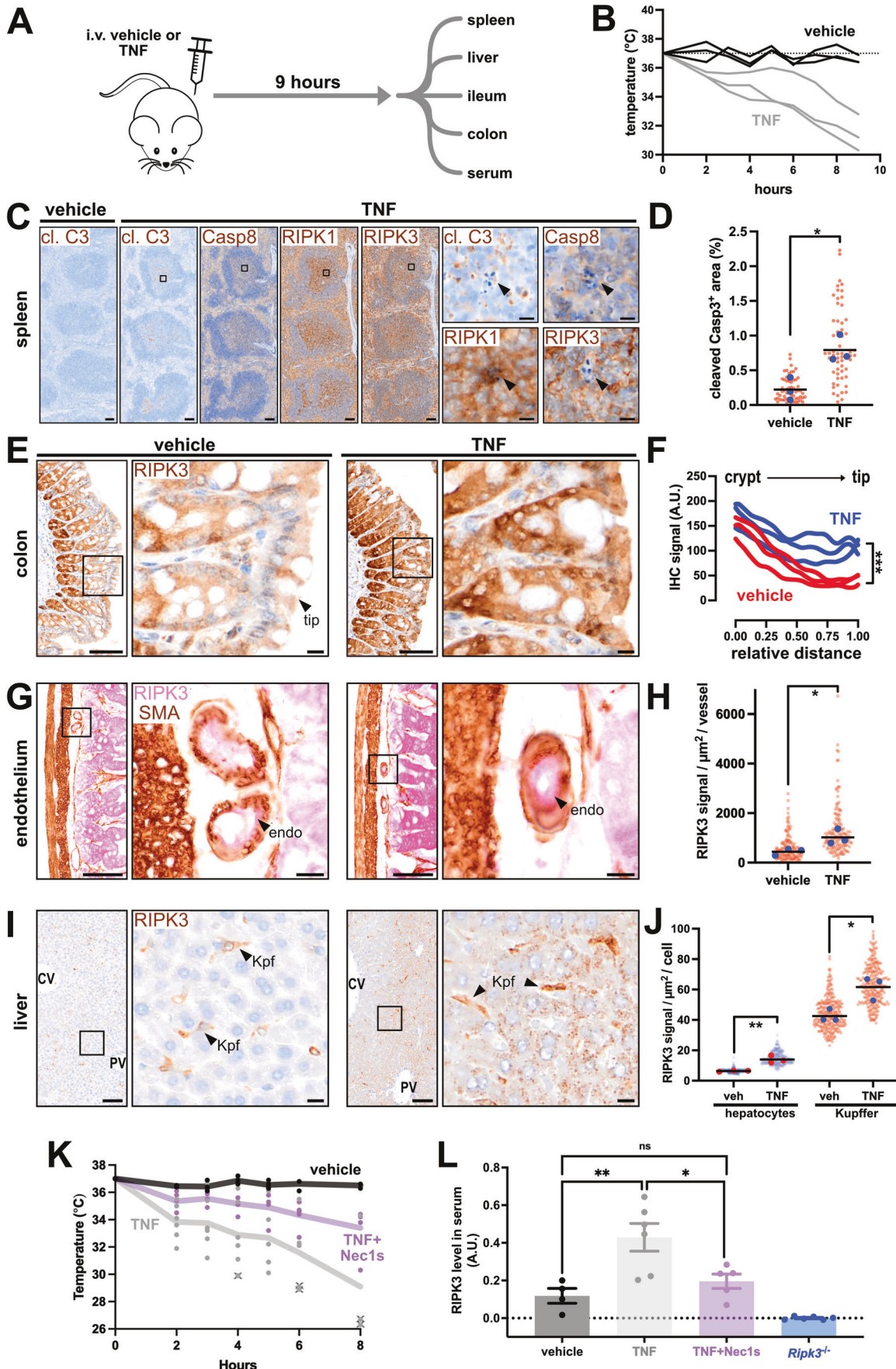

**Figure 3. RIPK3 expression is rapidly altered during systemic inflammation.**

(A) Experimental design. (B) Core temperatures of vehicle- and TNF-injected mice ($n = 3$ mice per group). (C) Immunosignals for cleaved Caspase-3 (cl. C3), Caspase-8, RIPK1, or RIPK3 from the spleen of vehicle- or TNF-injected mice. Insets show unidentified RIPK1[high] cells that associate with apoptotic bodies in the splenic white pulp. In panels (C, E, G, I), the scale bars in zoomed-out images correspond to 100 μm, and the scale bars in zoomed-in images correspond to 10 μm. (D) Graph of white pulp area occupied by cleaved Caspase-3[+] material in vehicle- and TNF-treated mice. Each red datapoint represents one white pulp lobule ($N = 20$ lobules/mouse). Blue datapoints indicate the median value per mouse ($n = 3$ mice/treatment). Black bars represent the mean value per group. *$p < 0.05$ by unpaired two-tailed $t$-test. (E) RIPK3 immunosignals in the colon of vehicle- or TNF-treated mice. (F) Best-fit curves of RIPK3 immunosignals along the crypt-to-tip axis from $N = 10$ axes per mouse ($n = 3$ mice/group). ***$p < 0.001$ by multiple unpaired two-tailed $t$-test. (G) RIPK3 (pink) and smooth muscle actin (brown) immunosignals in intestinal submucosa of vehicle- or TNF-treated mice. Insets show vessel cross-sections. Arrowheads show RIPK3[+] endothelial cells (endo). (H) Plot of RIPK3 signals per vessel. Each red datapoint represents one vessel ($N = 50$ vessels/mouse). Blue datapoints indicate the median value per mouse ($n = 3$ mice/treatment). Black bars represent the mean value per group. *$p < 0.05$ by unpaired two-tailed $t$-test. (I) RIPK3 immunosignals in the liver of vehicle- or TNF-treated mice. Central vein (CV), portal vein (PV) and Kupffer cell (Kpf). (J) Plot of RIPK3 signals per hepatocyte or Kupffer cell. Each transparent datapoint represents one cell ($N = 90$ cells/mouse). Opaque datapoints indicate the median value per mouse ($n = 3$ mice/treatment). Black bars represent the mean value per group. *$p < 0.05$ and **$p < 0.01$ by unpaired two-tailed $t$-test. (K) Core temperatures of vehicle-, TNF- and Nec1s+TNF-injected wild-type mice ($n = 4$–5 mice/treatment; one dot/mouse/time). Line indicates mean. X indicates a euthanized mouse due to its body temperature being <30 °C. (L) RIPK3 levels in serum from the mice in panel (K) or from untreated *Ripk3*[-/-] mice. Data expressed as arbitrary optical density units (A.U.). One dot per mouse. Mean ± SEM is shown. *$p < 0.05$, **$p < 0.01$ by one-way ANOVA with Tukey's post hoc correction. Source data are available online for this figure.

TNF-induced necroptosis (de Almagro et al, 2017; He et al, 2009; Li et al, 2021; Sun et al, 2012). Notably, the translocation of Caspase-8 and RIPK1, but not MLKL, to necrosomes could also be detected in mouse dermal fibroblasts undergoing TNF-induced necroptosis (Appendix Fig. S4). This species-dependent difference is likely due to dissimilarities in the interaction between RIPK3 and MLKL, which is thought to be more transient in mouse than in human cells (Petrie et al, 2018). Next, by combining automated immunohistochemistry with high-resolution digital slide scanning (~250 nm resolution) and customized image segmentation, we show that necrosomes can be detected and quantified across a large population of cells in an unbiased manner (Fig. 5B,C). We observed that the accuracy of segmenting human Caspase-8 or MLKL at necrosomes is higher than that of RIPK1, because the small puncta formed by necrosomal RIPK1 are near the resolution limit of existing brightfield slide scanners (Fig. 5C). Nonetheless, since necrosomes are a pathognomonic feature of necroptotic signaling, we propose that machine-based detection of necrosomes could be developed into a diagnostic assay for pinpointing necroptosis in formalin-fixed human patient biopsies. This proposal assumes that immunostaining protocols developed on cell pellets retain their specificity when applied to tissues. To test this assumption, we used two antibodies—one specific for mouse RIPK3 and one specific for human RIPK3—on spleens from *Ripk3*[-/-] mice, *Ripk3*[+/+] mice, or knock-in mice expressing human RIPK3 (see Methods and Protocols). As shown in Fig. EV3, immunoblotting and immunohistochemistry with the respective antibodies accurately discriminated between the expression of mouse RIPK3 or human RIPK3 in the spleen. These data suggest that immunohistochemistry protocols optimized on cell pellets can also be used to specifically stain tissues.

## Necrosome immunodetection in patients with IBD

Ulcerative colitis (UC) and Crohn's disease (CD) are the main types of IBD (Kobayashi et al, 2020; Roda et al, 2020). The causes of adult-onset IBD are multifactorial (Ananthakrishnan, 2015; Graham and Xavier, 2020). While many studies show that excess necroptosis promotes IBD-like pathology in mice (Gunther et al, 2011; Matsuzawa-Ishimoto et al, 2017; Schwarzer et al, 2020; Vlantis et al, 2016; Wang et al, 2020a; Xie et al, 2020; Xu et al, 2023), few studies have examined the prevalence of necroptosis in

IBD patients (Negroni et al, 2017; Pierdomenico et al, 2014; Shi et al, 2020). One Phase II trial of a RIPK1 inhibitor in UC has failed to demonstrate clinical efficacy (Weisel et al, 2021), but several other clinical and preclinical trials of RIPK1 inhibitors in IBD are underway. Thus, the role of necroptosis in IBD requires further investigation. We collected intestinal biopsies from adults with UC, CD, and non-IBD patients (Fig. 6A, together with clinical information in Appendix Table S1). To capture the chronology of disease, biopsies were collected from endoscopically "non-inflamed", "marginally inflamed", and "inflamed" intestinal tissue from patients with IBD. The grading of inflammation was verified by blinded histopathology scores (Fig. 6B). Cell death signaling in biopsies from each location and endoscopic grade was assessed by immunoblot (Fig. 6C; Appendix Table S1). To assist interpretation, patient samples (blue annotations in Fig. 6C) were immunoblotted alongside lysates from HT29 cells undergoing apoptosis or necroptosis (red annotations in Fig. 6C). Apoptotic signaling was inferred from increases in the conversion of pro-Caspase-3, -8, and -10 into their active cleaved forms (open arrowheads in Fig. 6C). Necroptotic signaling was inferred from increases in the abundance of phosphorylated RIPK3 and MLKL, relative to their non-phosphorylated forms (asterisks in Fig. 6C). This approach showed that cell death signaling is elevated in intestinal tissue from patients with IBD relative to patients without IBD, especially in inflamed intestinal biopsies from IBD patients (Figs. 6C and EV4). However, marked heterogeneity in the prevailing form of cell death was apparent in both UC and CD patients; with apoptosis dominant in some IBD cases (patients B and F in Fig. 6C and patient I in Fig. EV4), and necroptosis dominant in others (patients D and H in Fig. 6C). Given the ongoing development of RIPK1 inhibitors, it is noteworthy that phosphorylated MLKL coincided with phosphorylated RIPK1 in some, but not all patients with IBD (patients D versus H in Fig. 6C). Why cell death mechanisms vary between patients is currently unknown. Collectively, we find that cell death signaling increases in the inflamed gut, supporting the idea that cell death inhibitors are a potential treatment option for IBD. Whether apoptosis or necroptosis manifests in an individual IBD patient appears to be highly variable, highlighting the need for diagnostic approaches, such as automated immunohistochemistry, to identify patients who may benefit from anti-necroptotic therapy.

Having identified biopsies with differing modes of intestinal cell death, we next applied our panel of automated

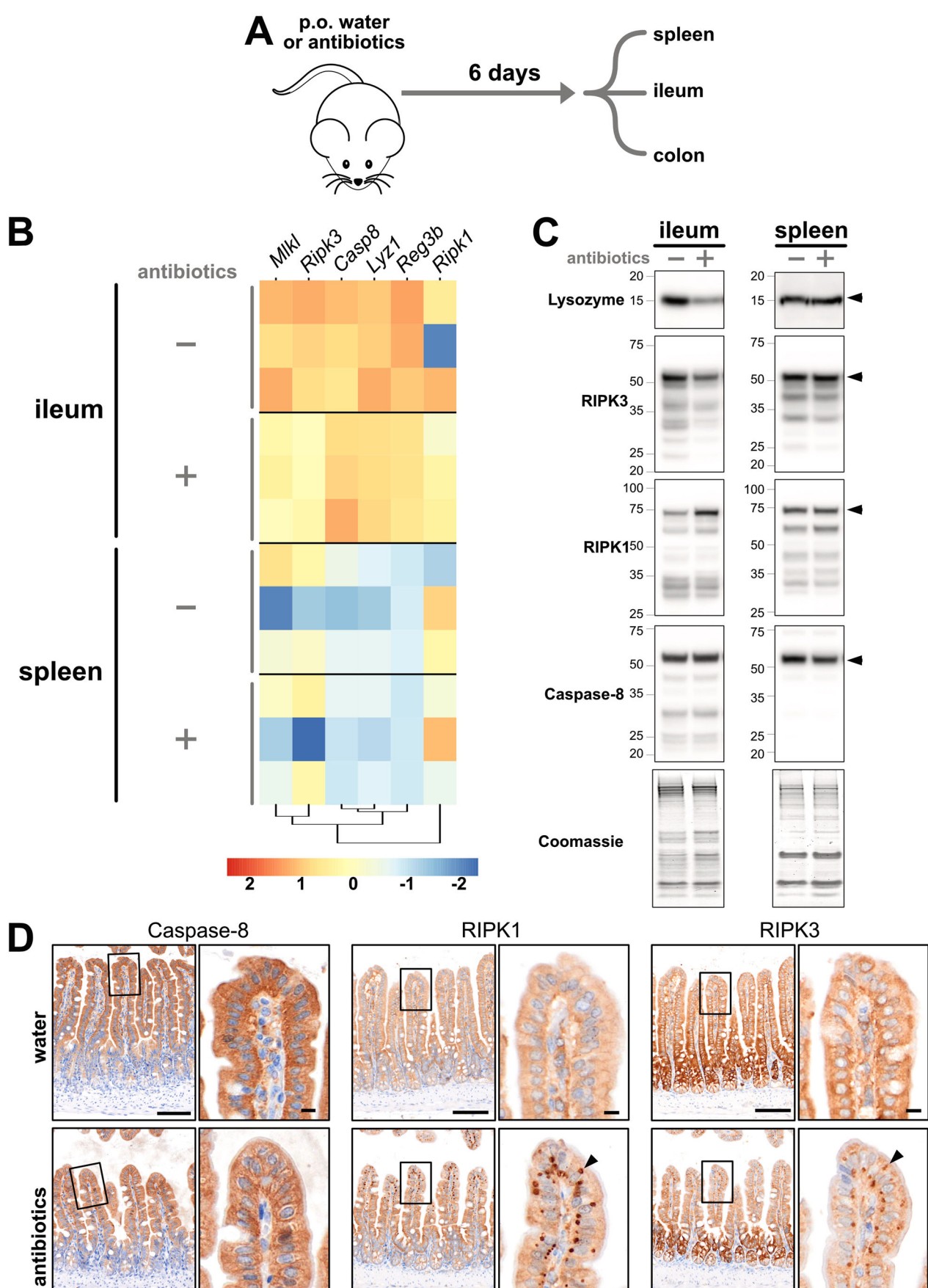

**Figure 4. RIPK3 expression changes in response to dysbiosis.**

(A) Experimental design. (B) Bulk RNA sequencing was performed on indicated tissues. Heatmap depicts the log-fold-change in gene expression for antibiotic- versus water-treated mice. Each row represents a different mouse. The legend shows the color-to-value scale. (C) Immunoblots for the indicated proteins in the ileum and spleen of water- versus antibiotic-treated mice. Arrowheads indicate full-length proteins of interest. Coomassie staining of total protein content was used as a loading control. Data were representative of $n = 7$ mice per tissue per group. (D) Caspase-8, RIPK3, and RIPK3 immunosignals in the ileum of water- or antibiotic-treated mice. Arrowheads to cytosolic accumulations of RIPK1 and RIPK3 in epithelial cells at villi tips. Scale bars in lower magnification micrographs are 100 μm. Scale bars in insets are 10 μm. Data were representative of $n = 7$ mice per group. Related to Appendix Fig. S2. Source data are available online for this figure.

immunohistochemical stains to matched biopsies (collected from the same patients at the same time and from the same location; Appendix Table S1; note that a biopsy for immunohistochemistry was not collected from patient F). Consistent with immunoblotting data showing increased apoptosis in patients B and I (Figs. 6C and EV4), immunohistochemistry detected substantially more cleaved Caspase-3[+] epithelial cells in patients B and I (open arrowheads Fig. EV5; Appendix Fig. S5) compared to other patients in the cohort. No obvious changes to epithelial RIPK1, RIPK3, and MLKL were evident between patients (Fig. EV5; Appendix Fig. S5). By comparison, cytoplasmic clusters of Caspase-8 were evident in the epithelial layer of the inflamed biopsy from patient D and H (closed arrowheads in Fig. 6D, closed arrowheads in EV5 and Appendix Fig. S5). Since these cytoplasmic clusters of Caspase-8 were reminiscent of the Caspase-8[+] necrosomes in necroptotic HT29 cells (Fig. 5), we used the same high-resolution digital slide scanning and unbiased image segmentation approach as before. This quantitation showed that the number of intraepithelial Caspase-8[+] clusters was low in non-IBD patient C and increased with inflammation in IBD patient D (Fig. 6E). This trend mirrors the levels of necroptotic signaling in IBD patient D (Fig. 6B), suggesting that cytoplasmic clusters of Caspase-8[+] may represent bona fide necrosomes. Why the immunohistochemical detection of Caspase-8 clusters distinguishes between necroptosis and apoptosis is unclear, but this likely relates to the observation that Caspase-8-containing complexes that form during necroptosis are qualitatively different from those that form during apoptosis (de Almagro et al, 2017). This case study suggests that automated immunohistochemical detection of Caspase-8[+] clusters in patient biopsies is feasible and could be developed into a diagnostic assay for pinpointing necroptosis in clinical practice. To this end, future studies with a larger number of biopsies are needed to determine whether the immunoblot detection of necroptotic signaling significantly correlates with the immunohistochemical detection of Caspase-8[+] clusters.

## Discussion

The difficulty of reliably detecting necroptotic signaling in fixed tissues has been a longstanding issue, generating confusion and conflicting results in the literature. To address this problem, we optimized the immunohistochemical detection of Caspase-8, RIPK1, RIPK3, and MLKL in formalin-fixed paraffin-embedded samples. While our prior studies immunostained non-crosslinked fixed monolayers (Samson et al, 2021a; Samson et al, 2020), here we used formalin-fixed paraffin-embedded specimens consistent with standard practice in clinical pathology and research departments around the world. In total, over 300 different immunostaining conditions were tested, yielding 13

automated immunohistochemistry protocols that we anticipate will be of broad utility to the cell death community and drive new insight into the causes, circumstances, and consequences of necroptosis. To assess the reliability of our automated protocols, we benchmarked our immunostaining results against data obtained using other methodologies. For instance, our automated immunohistochemistry protocols produced results that were comparable to public resources generated using proteomic, single-cell transcriptomic, and spatial transcriptomic approaches (Ben-Moshe et al, 2019; Geiger et al, 2013; Moor et al, 2018; Tabula Muris et al, 2018). Confidence was also taken from the similar staining patterns produced between three Caspase-8 antibodies (clones D53G2, 3B10, and 1G12), between two RIPK3 antibodies (clones 8G7 and 1H12), and between mouse and human intestinal tissue using the same RIPK1 antibody (clone D94C12). Thus, multiple lines of evidence suggest that the automated protocols described herein are specific and sensitive.

Our approach for optimizing and interpreting immunohistochemical signals relied upon quantitative analyses that carries important technical considerations. First and foremost, the signals produced by immunohistochemistry are non-linear (Bobrow and Moen, 2001). Moreover, before quantitation, we digitally unmixed immunosignals from the haematoxylin counterstain, which is another non-linear transformation of signal intensity (Landini et al, 2021). Thus, only relative changes in expression levels were inferred from changes in immunohistochemical signal intensity. Because of this caveat, we only compared and quantified immunosignals between closely matched specimens, such as corresponding wild-type and knockout samples, where both samples were sectioned at the same time, mounted on the same slide, and stained and imaged contemporaneously. To aid quantitation, all our immunohistochemistry protocols were developed to produce unsaturated signals. One final salient point is that this study used automated embedding and immunostaining procedures, with all quantification using macros that analyse a high number (typically thousands) of cells per sample. Thus, the automated immunohistochemistry protocols described herein can be used for quantitative purposes, but only when comparing closely matched specimens, and ideally with supporting data from alternative methodologies such as spatial transcriptomics. These recommendations reduce, but do not eliminate, the need to use knockout samples as a control for specificity.

The phosphorylated forms of RIPK1, RIPK3 and MLKL are the most widely-used markers of necroptotic signaling (Horne et al, 2023). Accordingly, prior attempts to detect necroptotic signaling in fixed specimens have focussed on phospho-RIPK1, -RIPK3, and -MLKL (Dominguez et al, 2021; He et al, 2021; Li et al, 2017; Li et al, 2022; Rodriguez et al, 2022; Rodriguez et al, 2016; Samson et al, 2021a; Samson et al, 2020; Wang et al, 2014; Webster et al, 2018; Zhang et al, 2020). However, there are drawbacks to this

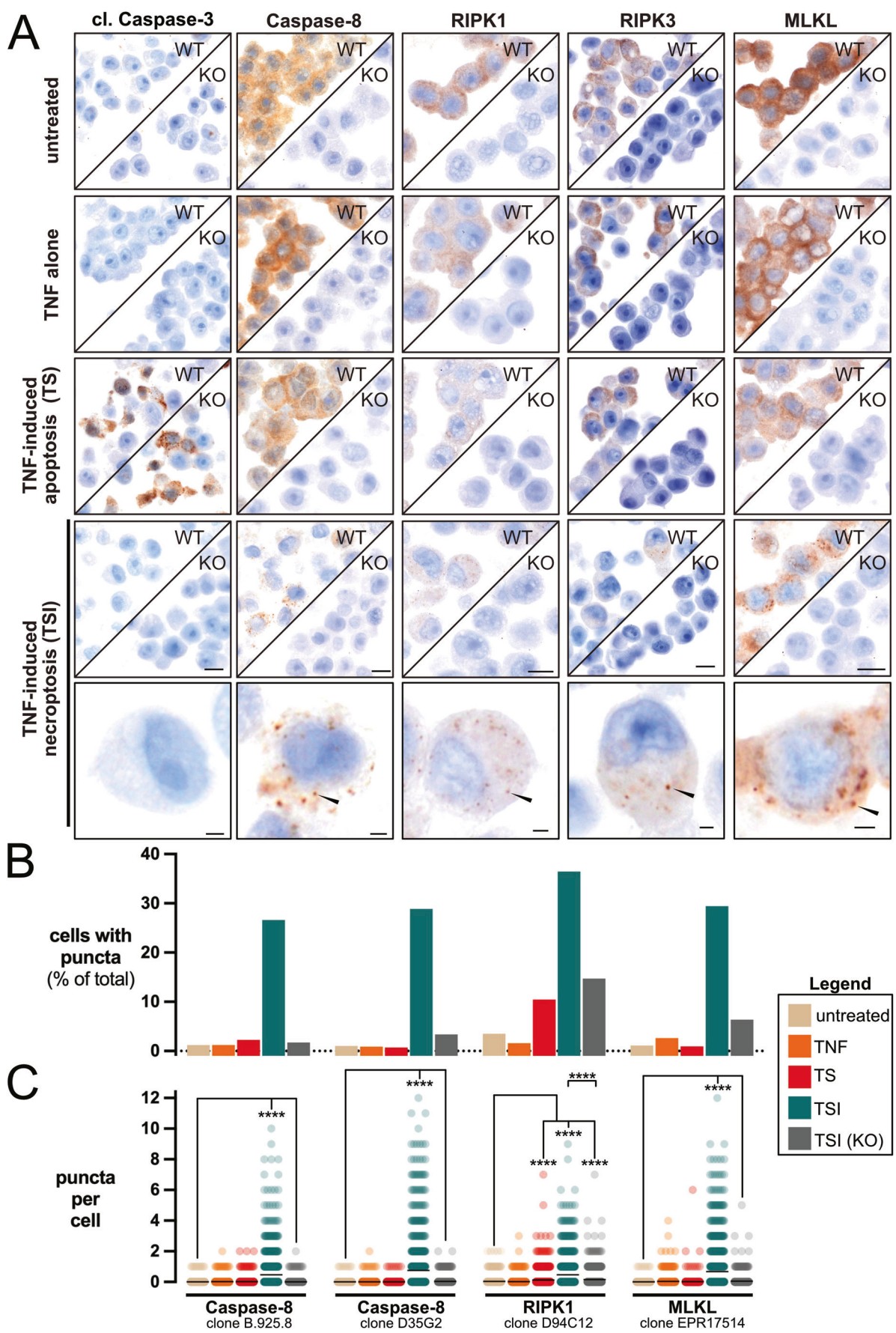

**Figure 5. Automated immunohistochemistry quantifies necroptotic signaling in human cells.**

(A) Immunosignals of cleaved Caspase-3, Caspase-8, RIPK1, RIPK3, and MLKL in wild-type versus *MLKL*[-/-] or *RIPK1*[-/-], or *CASP8*[-/-]*CASP10*[-/-]*MLKL*[-/-] HT29 cells. Arrowheads indicate Caspase-8[+], RIPK1[+], RIPK3[+], and MLKL[+] puncta that are presumed to be necrosomes. Data were representative of $n \geq 2$ for each protein and treatment. Scale bars in lower magnification micrographs are 10 μm. Scale bars in insets are 2 μm. (B) The percent of cells per treatment group that contain cytosolic necrosome-like puncta immunostained by the stipulated antibody. $N = 1051$–5630 cells were analysed per condition per stain. Data representative of $n = 2$ experiments. (C) The number of puncta per cell. $N = 1000$ cells per treatment group were analysed. Each datapoint represents one cell. The black bar indicates the mean value. ****$p < 0.0001$ by one-way ANOVA with Krukal–Wallis post hoc correction. Data representative of $n = 2$ experiments. Related to Appendix Figs. S3, S4 and EV3. Source data are available online for this figure.

approach: (1) antibodies against phosphorylated epitopes in RIPK1/3 and MLKL exhibit much poorer signal-to-noise properties than do antibodies against unphosphorylated epitopes in RIPK1/3 and MLKL (Samson et al, 2021a); (2) while knockout samples are sufficient for verifying non-phospho-signals, multiple controls are needed to authenticate phospho-signals (e.g., resting, knockout and phosphatase pre-treated samples); and (3) because only a small fraction of RIPK1/3 and MLKL is phosphorylated during necroptosis these phospho-species are inherently more difficult to detect than their unphosphorylated counterparts. Indeed, as is standard practice when detecting necroptosis via immunoblot, the immunohistochemical detection of phospho-RIPK1, -RIPK3, and -MLKL can only be interpreted when their unphosphorylated forms can also be detected. For these reasons, we focused on the immunohistochemical detection of unphosphorylated RIPK1, RIPK3, and MLKL, and used the immunodetection of necrosomes as a marker of necroptotic signaling.

It was surprising that expression of the necroptotic pathway was heavily restricted under steady-state conditions in mice. That fast-cycling progenitors and immune barrier cells are the dominant expressors of RIPK3 and/or MLKL suggests that the existential role of necroptosis is to protect the host from invading pathogens. The phenotypes and cell types affected in mice carrying activation-prone polymorphisms in *Mlkl* also supports the view that the ancestral role of necroptosis lies in innate immunity (Garnish et al, 2023; Hildebrand et al, 2020; Zhu et al, 2022). The absence of necroptotic pathway expression in slow-cycling cell populations was equally striking, with RIPK3 and/or MLKL undetectable in resident cells of the heart (except for certain fibroblasts), the brain (except for leptomeningeal vessels), the kidney and the liver (except for Kupffer cells) in mice under basal conditions. These observations challenge a vast body of literature, highlighting the need for robust well-controlled methodologies. These results lead us to propose that slow-cycling cell populations ensure longevity by avoiding inadvertent necroptotic signaling. The observation that RIPK3 expression is rapidly derepressed in hepatocytes during inflammation supports the notion that long-lived cells actively suppress necroptotic signaling, but only in the absence of challenge. The derepression of RIPK3 may reconcile the contribution of necroptosis in inflammatory disorders such as hepatocellular carcinoma (Vucur et al, 2023), acute myocardial infarction (Luedde et al, 2014), acute ischemic stroke (Degterev et al, 2005) and kidney ischemia-reperfusion injury (Linkermann et al, 2013). Notably, the Human Protein Atlas lacks immunohistochemical data for RIPK3 and MLKL, and therefore, it remains unknown whether the necroptotic pathway is similarly restricted in humans (Uhlen et al, 2015).

It is noteworthy that certain non-mitotic cells, such as bone marrow-derived macrophages, are highly sensitive to necrotic stimuli (Zelic and Kelliher, 2018). Thus, necroptosis is not reliant upon mitosis. Nonetheless, given the striking correlation between RIPK3 and mitotic marker expression, and since RIPK3's activity and interactome vary considerably during the cell cycle (Gupta and Liu, 2021; Liccardi et al, 2019), future studies should investigate how cell proliferation influences necroptotic susceptibility.

To exemplify the utility of our approach, we applied our full immunohistochemistry panel to determine whether inflammation, dysbiosis, or immunization alters necroptotic pathway expression in mice. Each of these challenges altered necroptotic pathway expression in a manner that chiefly involved local shifts in RIPK3 expression. This profiling of the necroptotic pathway yielded many unexpected observations, including: (1) RIPK3 expression is disinhibited in hepatocytes after TNF administration, with contemporaneous increases in circulating RIPK3 levels; (2) RIPK3 expression is suppressed in the gut during antibiotic administration with RIPK1/3 coalescing into unidentified cytoplasmic clusters in epithelial cells at the villus tip; and (3) RIPK3 expression is uniquely upregulated in splenic germinal centers after immunization. While the mechanistic basis for these findings warrants future attention, their initial description here illustrates the benefits of studying the necroptotic pathway using automated immunohistochemistry, and raises questions about whether RIPK3 expression is also regulated similarly in humans. This line of enquiry circles back to the idea that variations in RIPK3 expression are a key determinant of necroptotic potential (Cook et al, 2014; He et al, 2009; Najafov et al, 2018).

Our data raise the possibility that the immunodetection of intracellular Caspase-8[+] clusters necrosomes could be used as an in situ marker of necroptotic signaling in IBD patients. To reach this conclusion, we developed a suite of automated immunohistochemical protocols to detect the relocation of necroptotic effectors into necrosomes and validated the presence of necroptotic signaling in closely matched biopsies from patients with IBD using immunoblot. Collectively, these experiments show that intracellular clusters of Caspase-8, RIPK1, RIPK3, and MLKL are readily detectable under idealized cell culture conditions, that elevated necroptotic signaling occurs in a subset of IBD, and that cytoplasmic clustering of Caspase-8 correlated with necroptotic signaling across a set of biopsies from two patients with IBD. Why clusters of Caspase-8, but not clusters of RIPK1 or MLKL, were detectable in biopsies with active necroptotic signaling is unknown but may be due to technical limitations (e.g., resolution limit) or gaps in our understanding of how necroptosis manifests in vivo. Another important issue is whether the immunohistochemical detection of Caspase-8[+] clusters can be used to quantify necroptotic signaling in larger cohorts of patients with IBD and in patients with other clinical indications. Notwithstanding these issues, the

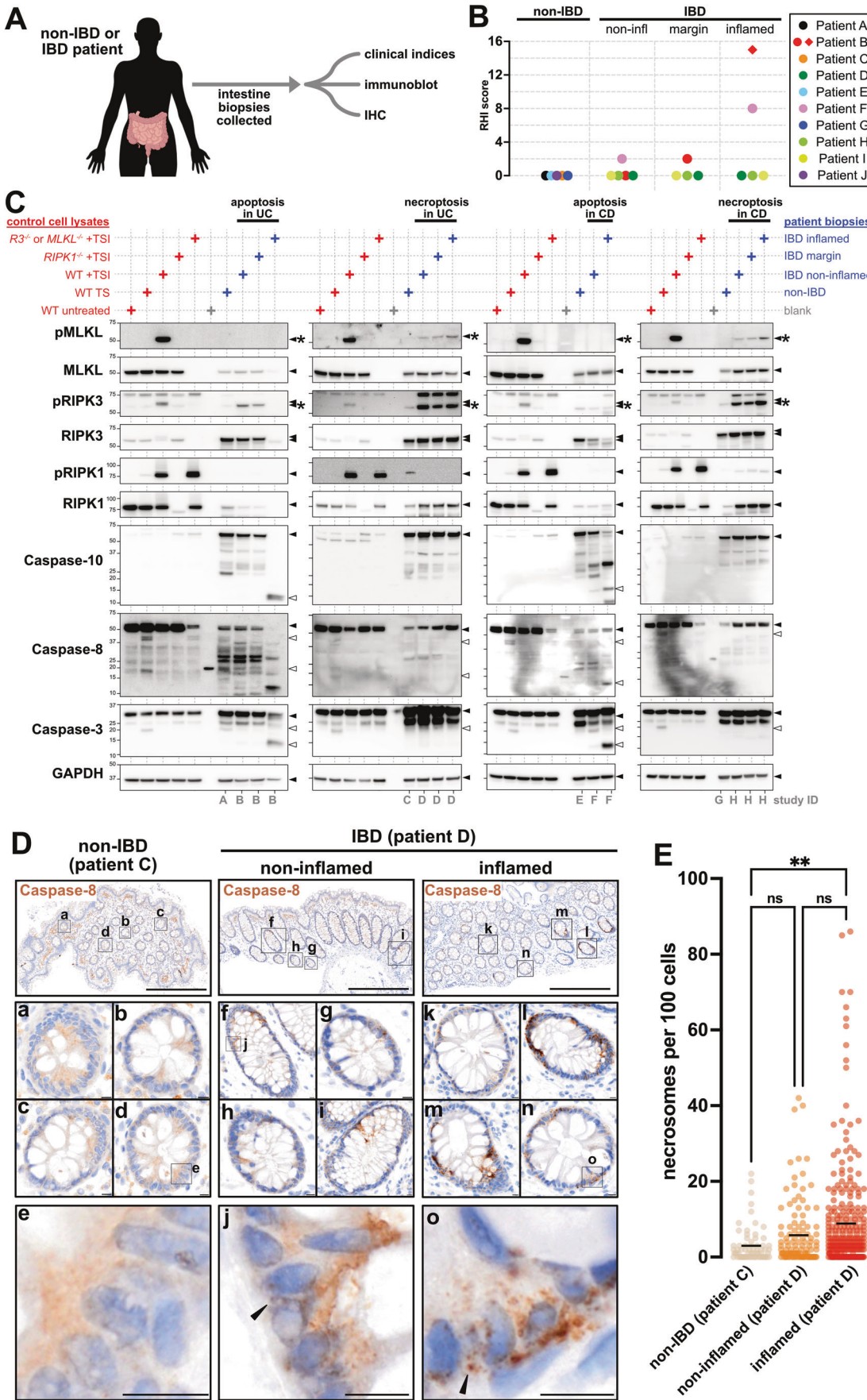

**Figure 6. Case study for the detection of necroptotic signaling in inflammatory bowel disease.**

(A) Study design. (B) Blinded histopathological (Robarts Histopathology Index; RHI) scores of disease activity in intestinal biopsies relative to their endoscopic grading of inflammation. Diamond indicates a sample that could not be formally scored, as it was solely comprised of neutrophilic exudate, but was given a pseudo-score of 15 that likely underrepresents the extent of disease activity in this biopsy. Biopsies scored in Panel B were matched to those used in panels (C–E) and Fig. EV4 (see Appendix Table S1 for details). (C) Immunoblot of lysates from HT29 cells (red annotations) and intestinal biopsies from patients A-H (blue annotations). The fifth lane of each gel contained lysates from TSI-treated *RIPK3$^{-/-}$* or TSI-treated *MLKL$^{-/-}$* cells (see source data for details). Patients A,C,E,G were non-IBD controls. Patients B and D had ulcerative colitis (UC). Patients F and H had Crohn's disease (CD). The endoscopic grading of the biopsy site as "non-inflamed", "marginally inflamed", or "inflamed" is stipulated. Closed arrowheads indicate full-length form of proteins. Asterisks indicate active, phosphorylated forms of RIPK3 (pRIPK3) and MLKL (pMLKL). Open arrowheads indicate active, cleaved forms of Caspase-8, Caspase-10, and Caspase-3. GAPDH was used as a loading control. (D) Immunohistochemistry for Caspase-8 (clone B.925.8) on intestinal biopsies. Insets a–e show diffuse epithelial Caspase-8 in patient C. Insets f–j show mild clustering of epithelial Caspase-8 and insets k–o show more pronounced clustering of epithelial Caspase-8 in patient D (arrowheads). Scale bars in lower magnification micrographs are 500 μm. Scale bars in insets are 10 μm. (E) The number of Caspase-8$^+$ puncta per 100 cells. Each datapoint represents one crypt. Whole slide scans with $N = 20{,}246$ cells from the 'non-IBD patient C' biopsy, $N = 10{,}416$ cells from the 'non-inflamed patient D' biopsy, and $N = 30{,}799$ cells from the 'inflamed patient D' biopsy were analysed. The black bar indicates the mean value. **$p < 0.01$ by one-way ANOVA with Tukey's post hoc correction. Related to Appendix Table S1; Figs. EV4, 5; Appendix Fig. S5. Source data are available online for this figure.

detection of in situ changes in necroptotic pathway expression in a scalable, quantitative, and automated manner represents a major leap forward in the capacity to pinpoint when and where necroptosis arises in both health and disease.

# Methods

### Reagents and tools table

| Reagent/Resource | Reference or source | Identifier or catalog number |
|---|---|---|
| **Experimental models** | | |
| Wild-type C57BL/6J (*M. musculus*) | Jackson Laboratory | Strain #000664; RRID: IMSR_JAX:000664 |
| *Mlkl$^{-/-}$* C57BL/6J (*M. musculus*) | (Murphy et al, 2013) | |
| *Ripk3$^{-/-}$* C57BL/6J (*M. musculus*) | (Tovey Crutchfield et al, 2023) | |
| *Ripk1$^{-/-}$ Ripk3$^{-/-}$ Casp8$^{-/-}$* C57BL/6J (*M. musculus*) | (Rickard et al, 2014) | Derived from strains reported by (Kelliher et al, 1998; Newton et al, 2004; Salmena et al, 2003) |
| *Ripk3$^{-/-}$ Casp8$^{-/-}$* C57BL/6J (*M. musculus*) | (Rickard et al, 2014) | Derived from strains reported by (Newton et al, 2004; Salmena et al, 2003) |
| *Ripk3$^{-/-}$ RIPK3$^{+/+}$* C57BL/6J (*M. musculus*) | Gifted by Anaxis Pharma Pty Ltd. | |
| HT29 human adenocarcinoma cell line | ATCC | Cat#HTB-38 |
| *RIPK1$^{-/-}$* HT29 cell line | (Tanzer et al, 2017) | |
| *RIPK3$^{-/-}$* HT29 cell line | (Garnish et al, 2021) | |
| *MLKL$^{-/-}$* HT29 cell line | (Petrie et al, 2018) | |
| *CASP8$^{-/-}$ CASP10$^{-/-}$ MLKL$^{-/-}$* HT29 cell line | (Tanzer et al, 2017) | |
| Wild-type mouse dermal fibroblasts | This study | Derived from wild-type C57BL/6J mice |

| Reagent/Resource | Reference or source | Identifier or catalog number |
|---|---|---|
| **Antibodies** | | |
| Rat anti-mouse Caspase-8 | In-house clone 3B10; available from Adipogen | Cat#AG-20T-0138 |
| Rat anti-mouse Caspase-8 | In-house clone 1G12; available from Adipogen | Cat#AG-20T-0137 |
| Rabbit anti-Caspase-8 | Cell Signaling Technology | Cat#4790 |
| Rabbit anti-phospho-RIPK1 | Cell Signaling Technology | Cat#44590 S |
| Mouse anti-RIPK1 | BD Biosciences | Cat#610459 |
| Mouse anti-RIPK1 | R&D Systems | Cat#MAB3585 |
| Rabbit anti-RIPK1 | Cell Signaling Technology | Cat#3493 |
| Rabbit anti-RIPK3 | Thermo Fisher Scientific | Cat#703750 |
| Rabbit anti-phospho-RIPK3 | Cell Signaling Technology | Cat#93654 |
| Rat anti-mouse RIPK3 | In-house clone 1H12; (Samson et al, 2021a) | |
| Rat anti-mouse RIPK3 | In-house clone 8G7; available from Millipore | Cat#MABC1595 |
| Rat anti-mouse RIPK3 biotin-conjugated | This study | Derived from in-house clone 8G7 (available as Millipore Cat#MABC1595) |
| Rabbit anti-phospho-MLKL | Cell Signaling Technology | Cat#37333 |
| Rabbit anti-phospho-MLKL | Abcam | Cat#ab187091 |
| Mouse anti-MLKL | ProteinTech | Cat#66675-1-IG |
| Rabbit anti-mouse MLKL | Cell Signaling Technology | Cat#37705 |
| Rat anti-mouse MLKL | In-house; available from Millipore | Cat#MABC1634 |
| Rat anti-MLKL | In-house; available from Millipore | Cat#MABC604 |

| Reagent/Resource | Reference or source | Identifier or catalog number |
|---|---|---|
| Mouse anti-Caspase-10 | MBL International | Cat#M059-3 |
| Mouse anti-Caspase-8 | Thermo Fisher Scientific | Cat#MA5-15226 |
| Mouse anti-Caspase-8 | MBL International | Cat#M058-3 |
| Rat anti-human RIPK3 | In-house clone 1H2; available from Millipore | Cat#MABC1640 |
| Rabbit anti-human RIPK3 | Cell Signaling Technology | Cat#13526 |
| Rabbit anti-human RIPK3 | Cell Signaling Technology | Cat#10188 |
| Rat anti-human MLKL | In-house clone 7G2; available from Millipore | Cat#MABC1636 |
| Rat anti-human MLKL | In-house clone 10C2; available from Millipore | Cat#MABC1635 |
| Rabbit anti-MLKL | Thermo Fisher Scientific | Cat#MA5-24846 |
| Rabbit anti-MLKL | Abcam | Cat#ab184718 |
| Mouse anti-GAPDH | Millipore | Cat#MAB374 |
| Rabbit anti-smooth muscle actin | Cell Signaling Technology | Cat#9245 S |
| Rabbit anti-lysozyme | Abcam | Cat#108508 |
| Rabbit anti-cleaved Caspase-3 | Cell Signaling Technology | Cat#9661 |
| Rabbit anti-Ki67 | Cell Signaling Technology | Cat#12202 |
| HRP-conjugated goat anti-rat Ig | Southern BioTech | Cat#3010-05 |
| HRP-conjugated goat anti-rabbit Ig | Southern BioTech | Cat#4010-05 |
| HRP-conjugated goat anti-mouse Ig | Southern BioTech | Cat#1010-05 |
| HRP-conjugated anti-rabbit Ig | Agilent | Cat#K400311-2 |
| HRP-conjugated anti-mouse Ig | Agilent | Cat#K400111-2 |
| HRP Goat Anti-Rat IgG Polymer Detection Kit | Vector Laboratories | Cat#MP740450 |
| HRP Goat Anti-Rat IgG | R&D Systems | Cat#VC005-125 |
| HRP-conjugated anti-digoxigenin | Cell Signaling Technology | Cat#49620 |
| Anti-mouse CD38 | In-house | Clone NIMR-5 |
| Rat anti-mouse CD19 | BD Biosciences | Cat#BD552854 |
| Anti-mouse IgM | In-house | Clone 331.12 |
| Anti-mouse IgD | In-house | Clone 11–26 C |
| Anti-mouse Gr-1 | In-house | Clone RB6-8C5 |
| Anti-mouse CD138 | BD Biosciences | Cat#BD564068 |
| Anti-mouse IgG1 | BD Biosciences | Cat#BD550874 |
| **Chemicals, enzymes, and other reagents** | | |

| Reagent/Resource | Reference or source | Identifier or catalog number |
|---|---|---|
| Epitope Retrieval Solution 1 | Leica | Cat #AR9961 |
| Epitope Retrieval Solution 2 | Leica | Cat #AR9940 |
| Retrieval Solution Low pH | Agilent | Cat#GV80511-2 |
| Retrieval Solution High pH | Agilent | Cat#GV80411-2 |
| DAB substrate | Agilent | Cat#GV82511-2 or GV92511-2 |
| Dako REAL Peroxidase-blocking reagent | Agilent | Cat#S202386-2 |
| Bluing reagent | Leica | Cat#3802915 |
| Background Sniper | Biocare Medical | Cat#BS966L |
| Dako Protein Block | Agilent | Cat#X0909 |
| 'Normal' block | Agilent | Cat#S202386-2 |
| EnVision FLEX TRS High pH | Agilent | Cat#GV80411-2 |
| EnVision FLEX TRS Low pH | Agilent | Cat#GV80511-2 |
| MACH4 universal HRP polymer | Biocare Medical | Cat#M4U534L |
| DPX | Trajan | Cat#EUKITT |
| Opal TSA-Dig | Akoya | Cat#OP-001007 |
| Opal Polaris 780 Reagent | Akoya | Cat#OP-001008 |
| Rabbit Linker | Agilent | Cat#GV80911-2 |
| Protease and phosphatase inhibitor cocktail | Cell Signaling Technology | Cat#5872 |
| Benzonase | Sigma-Aldrich | Cat#E1014 |
| 4–15% Tris-Glycine gel | Bio-Rad | Cat#5678084 |
| 4–12% Bis-Tris gel | Thermo Fisher Scientific | Cat#NP0335BOX |
| MES Running Buffer | Thermo Fisher Scientific | Cat#NP000202 |
| Enhanced chemiluminescence | Merck | Cat#WBLUF0100 |
| PVDF membrane | Merck | Cat#IPVH00010 |
| Coomassie stain | Thermo Fisher Scientific | Cat#LC6060 |
| Neutral buffered formalin | Australian Biostain | Cat#AGFG.5 L |
| TNF | R&D Systems | Cat#410-MT/CF Lot CS152103 |
| TNF | BioTimes | Cat#CF09-1MG |
| Endotoxin-free DPBS | Merck | Cat#TMS-012-A |
| Necrostatin-1s | Cell Signaling Technology | Cat#17802 |
| Streptavidin-HRP | SouthernBiotech | Cat#7100-05 |

| Reagent/Resource | Reference or source | Identifier or catalog number |
|---|---|---|
| 2'2-Azinobis (3-ethylbenzthiazoline Sulfonic Acid) diammonium salt | Sigma-Aldrich | Cat#A-1888 |
| Ampicillin | Sigma | Cat#PHR1424 |
| Neomycin | Sigma | Cat#PHR1491 |
| Metronidazole | Sigma | Cat#PHR1052 |
| Enrofloxacin | Sigma | Cat#PHR1513 |
| Meropenem | Sigma | Cat#32460 |
| Di-Vetelact supplement | Lillelund Pty Ltd. | |
| RNAlater | Thermo Fisher Scientific | Cat#4427575 |
| NucleoSPin RNAXS kit | Macherey-Nagel | Cat#SKU:740902.250 |
| Zirconium Beads | OPS Diagnostics | Cat#BAWZ 3000-300-23 |
| Qubit RNA HS Assay kit | Thermo Fisher Scientific | Cat#Q32852 |
| Qubit dsDNA Assay kit | Thermo Fisher Scientific | Cat#Q32851 |
| RNA ScreenTape | Agilent | Cat#5067-5579 |
| D1000 ScreenTape | Agilent | Cat#5067-5582 |
| SMARTer Stranded Total RNA-Seq Pico-Input Mammalian kit v.3 | Takara Bio | Cat#SKU:634487 |
| P2 300- cycle kit v3 | Illumina | Cat#20046813 |
| 4-hydroxy-3-nitrophenylacetyl coupled to keyhole limpet hemocyanin | In-house | |
| Human TNF-Fc | In-house; (Bossen et al, 2006) | |
| Compound A Smac mimetic | A gift from Tetralogic Pharmaceuticals | |
| IDN-6556 pan-Caspase inhibitor | A gift from Idun Pharmaceuticals | |
| HistoGel | Epredia | Cat#HG-4000-012 |
| DMEM high glucose | Thermo Fisher Scientific | Cat#11965092 |
| Fetal Calf Serum | Sigma | Cat#F9423-500ML |
| DPBS | Thermo Fisher Scientific | Cat#14190144 |
| Protease inhibitor cocktail | Merck | Cat#4693132001 |
| Phosphatase inhibitor cocktail | Merck | Cat#4606837001 |
| **Software** | | |
| Pannoramic SCAN 150 1.23 SP1 RTM | 3D Histech | |
| SlideViewer 2.8.178749 | | |
| Olympus VS200 ASW 3.41 | | |
| ImageJ 1.53t | (Schindelin et al, 2012) | |

| Reagent/Resource | Reference or source | Identifier or catalog number |
|---|---|---|
| Stereopy v0.12.0 | https://github.com/BGIResearch/stereopy | |
| Anndata v0.7.5 | https://github.com/scverse/anndata | |
| Scanpy v1.9.2 | https://github.com/scverse/scanpy | |
| Squidpy package v1.2.2 | https://github.com/scverse/squidpy | |
| Real Time Analysis v2.4.6 | | |
| bcl2fastq conversion software v2.15.0.4 | | |
| CASAVA v1.8.2 | | |
| Cutadapt v1.9 | (Martin, 2011) | |
| HISAT2 | (Kim et al, 2019) | |
| Limma v3.40.6 | (Law et al, 2014; Liao et al, 2014; Robinson and Oshlack, 2010) | |
| CellPose | (Stringer et al, 2021) | |
| FastPathology v1.1.2 | (Pedersen et al, 2021; Pettersen et al, 2021) | |
| QuPath v0.4.3 | (Pedersen et al, 2021) | |
| Prism v9.5.1 | GraphPad | |
| **Other** | | |
| example: Illumina NextSeq 500 | Illumina | |
| ChemiDoc Touch Imaging System | Bio-Rad | |
| Tissue-Tek VIP 6 AI Tissue Processor | Sakura Finetek | |
| Autostainer XL | Leica | Cat#ST5010 |
| Dako Omnis | Agilent | |
| Pannoramic Scan II | 3D Histech | |
| VS200 Research Slide Scanner | Olympus | |
| VersaMax ELISA microplate reader | Molecular Devices | |
| TapeStation 4200 | Agilent | Cat#G2991BA |
| NextSeq2000 | Illumina | |
| Radial Jaw biopsy forceps | Boston Scientific | |
| EVIS EXERA III endoscopes | Olympus | |

## Antibodies for immunohistochemistry and immunoblotting

Primary antibodies trialed are listed below with their stock concentrations. A range of working dilutions were trialed for the following antibodies, although no conditions could be optimized for specificity and intensity: rabbit anti-phospho-RIPK1 (clone D813A; RRID:AB_2799268; Cell Signaling Technology Cat#44590

S); mouse anti-RIPK1 (clone 38/RIP; RRID:AB_397831; 0.25 g/L BD Biosciences Cat#610459); mouse anti-RIPK1 (clone 334640; RRID:AB_2253447; 0.5 g/L; R&D Systems Cat#MAB3585); rabbit anti-RIPK3 (clone 18H1L23; RRID: AB_2866471; 0.5 g/L; Thermo Fisher Scientific Cat#703750); rabbit anti-phospho-RIPK3 (clone D6W2T; RRID:AB_2800206; Cell Signaling Technology Cat#93654); rabbit anti-phospho-MLKL (clone D6E3G; RRID:AB_2799112; Cell Signaling Technology Cat# 37333); rabbit anti-phospho-MLKL (clone EPR9514; RRID:AB_2619685; Abcam Cat#ab187091; (Wang et al, 2014)); mouse anti-MLKL (clone 3D4C6; RRID:AB_2882029; 1.957 g/L; ProteinTech Cat#66675-1-IG); rat anti-MLKL (clone 3H1; RRID:AB_2820284; 2 g/L produced in-house (Murphy et al, 2013) and available from Millipore Cat# MABC604); rabbit anti-mouse MLKL (clone D6W1K; RRID:AB_2799118; Cell Signaling Technology Cat#37705); rabbit anti-MLKL (clone 2B9; RRID:AB_2717284; 1 g/L; Thermo Fisher Scientific Cat#MA5-24846); mouse anti-Caspase-10 (clone 4C1; RRID:AB_590721; 1 g/L; MBL International Cat# M059-3); mouse anti-Caspase-8 (clone 5D3; RRID:AB_590761; 1 g/L; MBL International Cat#M058-3); rat anti-human RIPK3 (clone 1H2; RRID:AB_2940816; 2 g/L produced in-house (Petrie et al, 2019) and available from Millipore Cat# MABC1640); rabbit anti-human RIPK3 (clone E1Z1D; RRID:AB_2687467; Cell Signaling Technology Cat# 13526); rat anti-human MLKL (clone 7G2; RRID:AB_2940818; 2 g/L produced in-house (Samson et al, 2020) and available from Millipore Cat# MABC1636). Optimal dilutions were determined for the following primary antibodies with the corresponding Autostainer protocols established in this study presented in Appendix Table S2: rat anti-mouse Caspase-8 (clone 3B10; RRID:AB_2490519; 1 g/L produced in-house (O'Reilly et al, 2004) and available from AdipoGen Cat#AG-20T-0138), 1:200 working dilution; rat anti-mouse Caspase-8 (clone 1G12; RRID:AB_2490518; 1 g/L produced in-house (O'Reilly et al, 2004) and available from AdipoGen Cat#AG-20T-0137), 1:200 dilution; rabbit anti-Caspase-8 (clone D35G2; RRID:AB_10545768; Cell Signaling Technology Cat#4790), 1:200 dilution; rabbit anti-RIPK1 (clone D94C12; RRID:AB_2305314; Cell Signaling Technology Cat#3493), 1:200 dilution; rat anti-mouse RIPK3 (clone 1H12; 2 g/L produced in-house (Samson et al, 2021a)), 1:100 dilution; rat anti-mouse RIPK3 (clone 8G7; RRID: RRID:AB_2940810; 2 g/L produced in-house (Petrie et al, 2019) and available from Millipore Cat#MABC1595), 1:500 dilution; rat anti-mouse MLKL (clone 5A6; RRID:AB_2940800; 50 g/L produced in-house (Samson et al, 2021a) and available from Millipore Cat#MABC1634), 1:200 dilution; mouse anti-human Caspase-8 (clone B.925.8; RRID:AB_10978471; 0.619 g/L Thermo Fisher Scientific Cat# MA5-15226), 1:50; rabbit anti-human RIPK3 (clone E7A7F; RRID:AB_2904619; Cell Signaling Technology Cat# 10188), 1:100 dilution; rat anti-human MLKL (clone 10C2; RRID:AB_2940821; 2 g/L produced in-house (Samson et al, 2020) and available from Millipore Cat# MABC1635), 1:500 dilution; rabbit anti-human MLKL (clone EPR17514; RRID:AB_2755030; 1.9 g/L; Abcam Cat# ab184718), 1:500 dilution; rabbit anti-smooth muscle actin (clone D4K9N; RRID:AB_2734735; Cell Signaling Technology Cat#9245 S), 1:300 dilution; rabbit anti-cleaved-caspase-3 (Cell Signaling Technology, #9661), 1:300 dilution; rabbit anti-Ki67 (Cell Signaling Technology, #12202), 1:400 dilution. The concentration of antibodies from Cell Signaling Technology is often not provided and thus was not listed here. For immunoblots, rat primary antibodies were used at 1:2000 dilution and other primary antibodies at 1:1000 (as detailed below in Immunoblots). The following were used in immunoblots only: rabbit anti-lysozyme (clone EPR2994(2); RRID:AB_10861277; Abcam Cat#108508); mouse anti-GAPDH (clone 6C5; RRID:AB_2107445; 1 g/L; Millipore Cat# MAB374).

Secondary antibodies for immunoblotting (1:10,000 working concentration) were horseradish peroxidase (HRP)-conjugated goat anti-rat immunoglobulin (Ig) (Southern BioTech Cat#3010-05), HRP-conjugated goat anti-rabbit Ig (Southern BioTech Cat#4010-05), and HRP-conjugated goat anti-mouse Ig (Southern BioTech Cat#1010-05).

Reagents for immunohistochemistry were HRP-conjugated anti-rabbit Ig (Agilent Cat# K400311-2), HRP-conjugated anti-mouse Ig (Agilent Cat# K400111-2), ImmPRESS HRP-conjugated anti-rat IgG for human samples (Vector Laboratories Cat#MP740450), HRP-conjugated anti-rat IgG for mouse samples (R&D Systems Cat#VC005-125), HRP-conjugated anti-digoxigenin antibody (Cell Signaling Technology, clone D8Q9J, Cat#49620), Rabbit Linker (Agilent Cat#GV80911-2). Epitope Retrieval Solution 1 (Leica Cat#AR9961), Epitope Retrieval Solution 2 (Leica Cat#AR9640), Retrieval Solution Low pH (Agilent Cat#GV80511-2), Retrieval Solution High pH (Agilent Cat#GV80411-2), 3,3'-diaminobenzidine (DAB) substrate (Agilent Cat#GV82511-2 or GV92511-2), Dako REAL Peroxidase-blocking reagent (Agilent Cat#S202386-2), bluing reagent (Leica, Cat#3802915), Background Sniper (Biocare Medical Cat#BS966L), Dako Protein Block (Agilent Cat#X0909), 'Normal' block (Agilent Cat#S202386-2), EnVision FLEX TRS High pH (Agilent Cat# GV80411-2), EnVision FLEX TRS Low pH (Agilent Cat# GV80511-2), MACH4 universal HRP polymer (Biocare Medical Cat#M4U534L), and DPX (Trajan Cat#EUKITT), Opal TSA-Dig (Akoya Cat#OP-001007) and Opal Polaris 780 reagent (Akoya Cat#OP-001008).

## Mice, research ethics, and housing

All experiments were approved by The Walter and Eliza Hall Institute (WEHI) Animal Ethics Committee, Australia, in accordance with the Prevention of Cruelty to Animals Act (1986), with the Australian National Health and Medical Research Council Code of Practice for the Care and Use of Animals for Scientific Purposes (1997), and with the ARRIVE guidelines (Percie du Sert et al, 2020). Mice were housed at the WEHI animal facility under specific pathogen-free, temperature- and humidity-controlled conditions and subjected to a 12 h light/dark cycle with ad libitum feeding. Mice without functional MLKL alleles ($Mlkl^{-/-}$) have been described previously (Murphy et al, 2013). Mice without functional RIPK3 alleles ($Ripk3^{-/-}$) have been described previously (Tovey Crutchfield et al, 2023). Mice without functional alleles of RIPK1, RIPK3, and Caspase-8 ($Ripk1^{-/-}$ $Ripk3^{-/-}$ $Casp8^{-/-}$ triple knockout mice) and $Ripk3^{-/-}$ $Casp8^{-/-}$ double knockout mice have been described previously (Rickard et al, 2014) and were derived from reported mouse strains (Kelliher et al, 1998; Newton et al, 2004; Salmena et al, 2003). Human RIPK3 knock-in mice in which the human RIPK3 coding sequence was inserted into the mouse *Ripk3* locus of C57BL/6J mice were provided by Anaxis Pharma Pty Ltd.

## Mouse tissue lysate preparation

Mouse tissues were homogenized with a stainless steel ball bearing in a Qiagen TissueLyzer II (30 Hz, 1 min) in ice-cold RIPA buffer (10 mM Tris-HCl pH 8.0, 1 mM EGTA, 2 mM $MgCl_2$, 0.5% v/v

Triton X-100, 0.1% w/v sodium deoxycholate, 0.5% w/v sodium dodecyl sulfate (SDS), and 90 mM NaCl) supplemented with 1x Protease and Phosphatase Inhibitor Cocktail (Cell Signaling Technology Cat#5872) and 100 U/mL Benzonase (Sigma-Aldrich Cat#E1014). 1 mL of RIPA buffer per 25 mg of tissue was used for homogenization.

## Immunoblot

For Appendix Figs. S1, S3 and Fig. EV3, RIPA lysates were boiled for 10 min in Laemmli sample buffer (126 mM Tris-HCl, pH 8, 20% v/v glycerol, 4% w/v sodium dodecyl sulfate, 0.02% w/v bromophenol blue, 5% v/v 2-mercaptoethanol) and fractionated by 4–15% Tris-Glycine gel (Bio-Rad Cat#5678084) using Tris-Glycine running buffer (0.2 M Tris-HCl, 8% w/v SDS, 0.15 M glycine). After transfer onto nitrocellulose, membranes were blocked in 1% w/v bovine serum albumin (BSA; for Caspase-8 antibody clone B.925.8) or 5% w/v skim cow's milk powder (for all other antibodies) in TBS + T (50 mM Tris-HCl pH7.4, 0.15 M NaCl, 0.1 v/v Tween-20), probed with primary antibodies (1:2000 dilution for rat primary antibodies or 1:1000 for other primary antibodies in the above blocking buffers; supplemented with 0.01% w/v sodium azide; see Antibodies for immunohistochemistry and immunoblotting above) overnight at 4 °C, washed twice in TBS + T, probed with an appropriate HRP-conjugated secondary antibody (also above), washed four times in TBS + T and signals revealed by enhanced chemiluminescence (Merck Cat#WBLUF0100) on a ChemiDoc Touch Imaging System (Bio-Rad). Between probing with primary antibodies from the same species, membranes were incubated in stripping buffer (200 mM glycine pH 2.9, 1% w/v SDS, 0.5 mM TCEP) for 30 min at room temperature and then re-blocked.

For Figs. 4, 6, Appendix Fig. S2, and EV4, RIPA lysates were boiled for 10 min in Laemmli sample buffer (126 mM Tris-HCl, pH 8, 20% v/v glycerol, 4% w/v SDS, 0.02% w/v bromophenol blue, 5% v/v 2-mercaptoethanol) and fractionated by 4–12% Bis-Tris gel (Thermo Fisher Scientific Cat#NP0335BOX) using MES running buffer (Thermo Fisher Scientific Cat#NP000202). After transfer onto polyvinylidene difluoride (Merck Cat# IPVH00010), gels were Coomassie-stained as per manufacturer's instructions (Thermo Fisher Scientific Cat#LC6060) and membranes were blocked in 5% w/v skim cow's milk powder in TBS + T and then probed as above.

## Mouse tissue fixation

Tissues were immediately harvested after euthanasia and placed in 10% v/v Neutral Buffered Formalin (NBF) (Confix Green; Australian Biostain Cat#AGFG.5 L). A ratio of one part tissue to >10 parts formalin was used. Unless stipulated, tissues were incubated in formalin at room temperature for 24–72 h.

## Paraffin-embedding, microtomy, and immunostaining

Unless stipulated, formalin-fixed cells/tissues were paraffin-embedded using the standard 8 h auto-processing protocol of the Tissue-Tek VIP® 6 AI Tissue Processor (Sakura Finetek USA). Paraffin-embedded cells/tissues were cut in 4µm-thick sections onto adhesive slides (Menzel Gläser Superfrost PLUS). Haematoxylin and eosin staining was done on the Autostainer XL (Leica ST5010). Immunohistochemistry was performed on an automated

system: the Bond RX (Leica) or the DAKO OMNIS (Agilent). While the automated immunohistochemistry protocols used in this manuscript are fully detailed in Appendix Table S2, a brief overview of these methods is as follows: Step 1 was deparaffinisation (Phase 1: Clearify Clearing Agent; Phase 2: deonised water), Step 2 was heat-induced antigen retrieval (EnVision FLEX TRS, High pH or Low pH retrieval buffer. Treatment time was antibody-dependent), Step 3 was endogenous peroxidase blocking (Dako REAL Peroxidase-blocking reagent. Incubation time was antibody-dependent), Step 4 was protein blocking (Background Sniper, Dako Protein Block or Normal block. Incubation 10 min), Step 5 was primary antibody incubation (dilutions and incubation times were antibody-dependent), Step 6 was option signal amplification (amplification technique was antibody-dependent. Rabbit linker. Incubation 15 min), Step 7 was secondary reagent with horseradish peroxidase (anti-rabbit HRP polymer, anti-mouse HRP polymer, MACH4 universal HRP polymer, ImmPRESS anti-rat HRP polymer, anti-rat HRP polymer and in-house HRP signal amplification detection. Reagent and incubation was antibody-dependent), Step 8 was signal detection (chromogen-substrate DAB. Onboard mixing and incubation 10 min), Step 9 was counterstaining(in-house made Mayer's Haematoxylin for 1 min followed by Bluing reagent for 1 min), Step 10 was dehydration, mounting in DPX and coverslipping using Leica CV5030 platform.

## Image acquisition

Unless stipulated, slides were scanned on: (1) 3D Histech Pannoramic Scan II (objective: magnification 20x, numerical aperture 0.8, media dry; software: Pannoramic SCAN 150 1.23 SP1 RTM and SlideViewer 2.8.178749) or (2) Olympus VS200 (objective: 20x, numerical aperture 0.8, media dry; software: Olympus VS200 ASW 3.41). Where higher resolution was required, slides were scanned on the Olympus VS200 using the 60x objective (numerical aperture 1.42, media oil).

## Post-acquisition processing of displayed micrographs

All displayed micrographs were acquired with VS200 Research Slide Scanner (Olympus). Representative full-resolution 8-bit RGB micrographs of the WT and KO tissues/cells were imported into ImageJ 1.53t (Schindelin et al, 2012). Brightness-and-contrast was adjusted to 0–235 units and then gamma levels adjusted by 1–2.5-fold. Capture settings and post-acquisition image transformations were held constant between any micrographs that were being compared.

## Immunohistochemistry signal-to-noise ratio (S/N) analysis

Sections of wild-type (WT) and corresponding knockout (KO) tissues/cells on the same slide were immunostained and imaged. Representative full-resolution 8-bit RGB micrographs of the WT and KO tissues/cells were imported into ImageJ 1.53t (Schindelin et al, 2012) and the "H-DAB" function of the "Color Deconvolution 2" plugin (Landini et al, 2021) was used to unmix the DAB and hematoxylin channels. A dilated mask of the auto-thresholded hematoxylin channel was applied to the corresponding DAB channel to select an area-of-interest, and then a histogram of pixel

intensities from the DAB channel for both the WT and KO micrographs was determined. The "WT histogram" was then divided by the "KO histogram" to yield a S/N curve (see Fig. 1A for example). A weighted integral of the S/N curve was calculated as a numerical index of specificity (red box in Fig. 1A). The representative micrographs used to calculate S/N curves were of equivalent size, included >1000 cells per sample and their gamma levels unchanged.

## Quantitation of zonation via immunohistochemistry

Representative full-resolution 8-bit RGB micrographs of the WT tissues were imported into ImageJ 1.53t (Schindelin et al, 2012). Brightness-and-contrast was adjusted to 0–235 units and the "H-DAB" function of the "Color Deconvolution 2" plugin (Landini et al, 2021) was used to unmix the DAB and hematoxylin channels. The look-up-table for the DAB channel was converted to grayscale and pixel values inverted. The "Segmented Line tool" was used to draw a line (27.4 μm wide to approximate 2 cell widths) along the main axis of the following zones: (1) crypt base to villi tip in the ileum, (2) crypt base to crypt tip in the colon, (3) pericentral hepatocytes to peri-portal hepatocytes, and (4) white matter to red pulp in the spleen. The "Plot Profile tool" was used to quantify the immunosignal along the drawn axis, which was then averaged across $N = 10–20$ representative zones per mouse.

## Quantitation of endothelial RIPK3 levels via immunohistochemistry

Sections were subjected to automated immunohistochemistry for smooth muscle actin (SMA) and RIPK3 (see Appendix Table S2). SMA signals were detected with a brown DAB product. RIPK3 signals were detected with a pink DAB product. Sections were not counterstained with hematoxylin. Full-resolution 8-bit RGB micrographs (taken with 60x objective; Olympus VS200) were imported into ImageJ 1.53t (Schindelin et al, 2012) and endothelial RIPK3 levels analysed using a custom semi-automated macro in a non-blinded manner. In brief, cross-sections of individual SMA$^+$ vessels were chosen, SMA and RIPK3 signals were unmixed using the "Color Deconvolution 2" plugin (Landini et al, 2021). An auto-thresholded mask of the SMA$^+$ intima was then used to segment the endothelial area. The total RIPK3 signal in this endothelial area was expressed per unit area. This procedure was repeated for $N = 50$ vessels per mouse.

## Quantitation of hepatocyte RIPK3 levels via immunohistochemistry

Sections were subjected to automated immunohistochemistry for RIPK3 (see Appendix Table S2). RIPK3 signals were detected with brown DAB product and sections were counterstained with hematoxylin. Representative full-resolution 8-bit RGB micrographs (taken with the 60x objective; Olympus VS200) were imported into ImageJ 1.53t (Schindelin et al, 2012) and RIPK3 levels within individual hepatocytes analysed using a custom fully-automated macro. In brief, haematoxylin-stained nuclei were unmixed from the RIPK3 signal using the "Color Deconvolution 2" plugin (Landini et al, 2021). Nuclei were segmented using the "Analyze Particles" function of ImageJ and hepatocyte nuclei were

distinguished from Kupffer cell nuclei on the basis of their larger size and circularity. The total RIPK3 signal in the cytosolic area surrounding hepatocyte nuclei was segmented, measured, and expressed per unit area. $N = 90$ hepatocytes per mouse were measured.

## Quantitation of Kupffer cell RIPK3 levels via immunohistochemistry

Sections were subjected to automated immunohistochemistry for RIPK3 (see Appendix Table S2). RIPK3 signals were detected with brown DAB product and sections counterstained with hematoxylin. Representative full-resolution 8-bit RGB micrographs (taken with the 60x objective; Olympus VS200) were imported into ImageJ 1.53t (Schindelin et al, 2012) and RIPK3 levels within individual Kupffer cells were analysed using a custom fully-automated macro. In brief, the RIPK3 signal was unmixed from Hematoxylin using the "Color Deconvolution 2" plugin (Landini et al, 2021). An auto-thresholded mask of cells with relatively high expression of RIPK3 was created. This mask, together with the comparatively small and non-circular shape of Kupffer cells, was used to segment individual Kupffer cells. The total RIPK3 signal within each Kupffer cell was then measured and expressed per unit area. $N = 90$ Kupffer cells per mouse were measured.

## Quantitation of apoptosis in splenic white pulp via immunohistochemistry

Sections were subjected to automated immunohistochemistry for cleaved Caspase-3 (see Appendix Table S2). Cleaved Caspase-3 signals were detected with a brown DAB product and sections counterstained with hematoxylin. Representative full-resolution 8-bit RGB micrographs (taken with the 60x objective; Olympus VS200) were imported into ImageJ 1.53t (Schindelin et al, 2012) and percent area with cleaved Caspase-3 signal was analysed using a custom fully-automated macro. In brief, the cleaved Caspase-3 signal was unmixed from Hematoxylin using the "Color Deconvolution 2" plugin (Landini et al, 2021). A predefined threshold (50–255 units) was applied, and its area expressed as a percentage of the total white pulp area. $N = 20$ white pulp lobules per mouse were measured.

## Systemic inflammatory response syndrome (SIRS)

For experiments in Fig. 3A–J, co-housed 8-week-old female C57BL/6J wild-type mice were administered either 300 μg/kg TNF (R&D Systems Cat#410-MT/CF Lot CS152103) in endotoxin-free DPBS or endotoxin-free DPBS alone (Merck Cat# TMS-012-A) via bolus tail vein injection. Core body temperature was measured hourly with a rectal probe. Mice were euthanized via carbon dioxide inhalation 9 h after injection.

For experiments in Fig. 3K,L, co-housed 6-month-old female C57BL/6J wild-type mice were administered either: (1) 6 mg/kg of Necrostatin-1s (Cell Signaling Technology Cat#17802) in dimethyl sulfoxide via intraperitoneal injection then 15–30 min later given 150 μg/kg TNF (BioTimes Inc. Cat#CF09-1MG), or 150 μg/kg TNF (R&D Systems Cat#410-MT/CF Lot CS152103) in DPBS, or DPBS alone via bolus tail vein injection. Core body temperature was measured hourly with a rectal probe. Mice were euthanized via

carbon dioxide inhalation 8 h after injection or if their temperature dropped below 30 °C.

## Enzyme-linked immunosorbent assay (ELISA) for mouse RIPK3

96-well ELISA plates (Sigma-Aldrich, Cat#CLS3795) were coated with rat anti-RIPK3 (clone 1H12; 5 µg/ml diluted in PBS) and incubated overnight at room temperature in humid conditions. Plates were then washed in PBS + 0.005% v/v Tween-20, then PBS, then distilled water. Mouse serum samples were diluted and titrated on the plate in block solution (PBS containing 1% v/v FCS, 0.002% v/v Tween-20, and 0.6% w/v skim milk powder). Plates were incubated at room temperature in humid conditions overnight. The plates were washed as before. Detection antibody, biotin-conjugated rat anti-RIPK3 (clone 8G7; 5 µg/ml diluted in block solution) was added to each well. Plates were incubated for 4 h at room temperature in humid conditions. Plates were washed as before. Streptavidin-HRP (SouthernBiotech, Cat#7100-05) was diluted in a block solution and added to each well. Plates were incubated for 1 h at room temperature in humid conditions. Plates were washed as before, and ABTS substrate solution (water containing 0.54 mg/mL w/v 2'2-Azinobis (3-ethylbenzthiazoline Sulfonic Acid) diammonium salt (Sigma-Aldrich, Cat#A-1888), and 0.1 M citric acid and 0.03% v/v hydrogen peroxide ($H_2O_2$)) was added to each well. Plates were left to develop for 30–45 min at room temperature protected from light. Color development was analysed on VersaMax ELISA microplate reader (Molecular Devices) using wavelengths $\lambda = 415$ nm minus $\lambda = 492$ nm.

## *Tabula muris* analysis

Robject files of the FACS-based T*abula Muris* (Tabula Muris et al, 2018) dataset were downloaded from Figshare. Expression values were normalized and analysed using Seurat v4.3.0 (Butler et al, 2018; Hao et al, 2021; Satija et al, 2015; Stuart et al, 2019). Within each cell ontology and tissue origin, the percentage of cells with expression values >0 for *Mlkl, Ripk3, Ripk1, Casp8, Mki67, and Top2a* was tabulated and colorised using Excel v16.74 (Microsoft).

## Spatial transcriptomics of mouse spleen

To boost germinal center numbers and size, one adult C57BL/6J mouse was infected intravenously with $1 \times 10^5$ *Plasmodium berghei* parasitised red blood cells, and then drug-cured at the onset of disease symptoms as described in (Ly et al, 2019). Twelve days later the mouse was euthanised, spleen dissected, fixed in 10% v/v Neutral Buffered Formalin and paraffin-embedded (as above). Sections of formalin-fixed paraffin-embedded spleen were cut onto slides, then spatial enhanced resolution omics-sequencing (stereo-seq) data were generated using pre-release chemistry and MGI sequencers as in (Chen et al, 2022a) by BGI, China. The spot-to-spot distance is 500 nm and the data were binned to $50 \times 50$ spots (25 µm²). Binning was performed in stereopy (https://github.com/BGIResearch/stereopy) before being converted to an anndata file (https://github.com/scverse/anndata). The stereo-seq data were first loaded and pre-processed using the standard Scanpy workflow (https://github.com/scverse/scanpy), and then principal component analysis and Leiden clustering was performed at the bin ($50 \times 50$)

level. Leiden clusters were then plotted using Uniform Manifold Approximation and Projection (UMAP) and on spatial coordinates using the Squidpy package (https://github.com/scverse/squidpy). Scanpy's 'rank_genes_groups' method (https://scanpy-tutorials.readthedocs.io) was used to generate a matrix of gene by cluster populated by corresponding log-fold changes from 1 versus all *t*-tests. For each zone/cell type of interest a matrix is constructed by subsetting the full gene by cluster matrix generated from 'rank_genes_groups' to just the specific genes of interest. These matrices were reduced to a single column sum aggregated vector. From the vectors of scores for each 'zone' of the spleen (White Pulp, Red Pulp, Germinal Centers, and Marginal Zones), the top five scores within the 50th or 75th (depending on expected transcriptional variance within zone) percentile of the maximum are selected and the corresponding Leiden clusters are aggregated hierarchically. After this aggregation of clusters, the "rank_genes_groups" method is rerun to calculate a new set of gene rankings and log-fold changes for zones rather than clusters. The aggregated scores for genes of interest in each group are calculated for each zone. These scores are the log-fold changes for each gene within each zone. Scores for the *Casp8, Ripk1, Ripk3,* and *Mlkl* genes are then extracted from the matrices for each "zone" of the spleen, and each individual cluster for side-by-side comparison in heatmaps. Software version used were Anndata (v0.7.5), Stereopy (v0.12.0), Scanpy (v1.9.2), Squidpy (v1.2.2), Numpy (v1.21.6), Pandas (v1.5.3), Matplotlib (v3.5.2), and Seaborn (v0.12).

## Antibiotic administration

Co-housed littermates were split across two cages. The drinking water for one cage was supplemented with 1 g/L ampicillin, 1 g/L neomycin, 1 g/L metronidazole, 0.5 g/L enrofloxacin, 2.5 g/L meropenem as in (Bader et al, 2023). Antibiotic-supplemented water was replaced after 3 days. To minimize weight loss, both cages were given ad libitum access to Di-Vetelact supplement (Lillelund Pty Ltd). Mice were euthanised after 6 days of treatment.

## RNA extraction, library preparation, and sequencing

Immediately following dissection, a ~2 mm³ piece of tissue was placed in 0.5 mL RNA*later* (Thermo Fisher Scientific Cat#4427575) then stored at −80 °C. Samples were thawed, RNA*later* was removed then tissues were transferred into screw-capped tube pre-filled with 350 µl of RA1 buffer of NucleoSPin RNAXS kit (Macherey-Nagel Cat# SKU: 740902.250). Tissues were homogenized with 10 pcs of 3 mm Acid-Washed Zirconium Beads (OPS Diagnostics Cat# BAWZ 3000-300-23) in a Qiagen TissueLyzer II (30 Hz, 5 min). Homogenized samples were centrifuged (1 min, $11,000 \times g$) to remove tissue debris then RNA was purified using Nucleospin RNAXS column kit as per manufacturer's instructions without adding a carrier RNA. The purified RNA was quantified using Qubit™ RNA HS Assay kit (Thermo Fisher Scientific Cat#Q32852) and RNA integrity was visualized in high sensitivity RNA ScreenTape (Agilent Cat# 5067- 5579) using TapeStation 4200 (Agilent Cat# G2991BA). Ten nanograms of RNA were used for preparing indexed libraries using SMARTer Stranded Total RNA-Seq Pico-Input Mammalian kit v.3 (Takara Bio. Cat# SKU: 634487) as per manufacturer's instructions (except that fragmentation was performed for 3 min of fragmentation at 94 °C and 13

cycles was used for PCR2). The library concentration was quantified by Qubit™ dsDNA Assay kit (Thermo Fisher Cat#Q32851) and library size was determined using D1000 ScreenTape (Agilent Cat# 5067-5582) and visualized in TapeStation 4200. Equimolar amounts of the libraries were pooled and diluted to 750pM for 150-bp paired-end sequencing on a NextSeq2000 instrument (Illumina) using the P2 300- cycle kit v3 chemistry (Illumina Cat# 20046813) as per the manufacturer's instructions. To produce the sequences, the base calling and quality scoring was performed by the real time analysis (v2.4.6) software. The FASTQ file generation and de-multiplexing for the samples was performed by the bcl2fastq conversion software (v2.15.0.4).

## Murine bulk RNA sequence analysis

The single-end 75 bp were demultiplexed using CASAVA v1.8.2 and Cutadapt (v1.9) was used for read trimming (Martin, 2011). The trimmed reads were subsequently mapped to the mouse genome (mm10) using HISAT2 (Kim et al, 2019). FeatureCounts from the Rsubread package (version 1.34.7) was used for read counting after which genes <2 counts per million reads (CPM) in at least three samples were excluded from downstream analysis (Liao et al, 2014, 2019). Count data were normalized using the trimmed mean of M-values (TMM) method and differential gene expression analysis was performed using the limma-voom pipeline (limma version 3.40.6) (Law et al, 2014; Liao et al, 2014; Robinson and Oshlack, 2010). Adjustment for multiple testing was performed per comparison using the false discovery rate (FDR) method (Benjamini and Hochberg, 1995). Heatmaps of logCPM were generated using pheatmap.

## NP-KLH immunization and analysis

Immunization was performed as previously described (Kong et al, 2022). 8–10-week-old $Ripk3^{+/+}$ or $Ripk3^{-/-}$ littermate mice received a single intraperitoneal injection of 100 µg 4-hydroxy-3-nitrophenylacetyl hapten coupled to keyhole limpet hemocyanin (NP-KLH; produced in-house) at a ratio of 21:1 with alum. Fourteen days after immunization mice were euthanized via $CO_2$ inhalation and spleens harvested. To determine immune response to NP immunization, single-cell splenic suspensions were stained as described using antibodies to the following surface molecules: CD38 (clone:NIMR-5, in-house), CD19 (clone:1D3, cat #BD552854), IgM (clone:331.12, in-house), IgD (clone:11–26 C, in-house), Gr-1 (clone:RB6-8C5, in-house), CD138 (clone:281.2, cat #BD564068) and IgG1 (clone:X56, cat #BD550874). NP-binding was detected as described (Smith et al, 2000).

## Mouse serum analyses

Blood was collected via cardiac puncture and immediately transferred to an EDTA-coated tube (Sarstedt AG Cat#20.1341). Blood was transferred into a clot activator tube (Sarstedt AG Cat#20.1344) and serum prepared as per the manufacturer's instructions.

## Human research ethics

Ethical approval for intestinal tissue collection from participants undergoing endoscopy procedures through the Gastroenterology Department at the Royal Melbourne Hospital was attained from the Human Research Ethics Committee (HREC): HREC 2021.074. This was in accordance with the National Health and Medical Research Council (NHMRC) National Statement on Ethical Conduct in Human Research (2008) and the Note for Guidance on Good Clinical Practice (CPMP/ICH-135/95). Site-specific governance was sought for each of the collaborating sites: WEHI and the University of Melbourne. Collaboration amongst all three participating institutions was officiated through the Melbourne Academic Center for Health Research Collaboration Agreement (Non-Commercial). The human research in this study was performed in accordance with the principles expressed in the WMA Declaration of Helsinki and conforms the principles set out in the Department of Health and Human Services Belmont Report. The human materials used in this study were obtained with informed consent from all subjects.

## Human intestinal biopsy collection

Adult patients with or without IBD scheduled for endoscopic evaluation of the lower gastrointestinal tract (flexible sigmoidoscopy or colonoscopy) at the Gastroenterology Department at the Royal Melbourne Hospital were screened for study inclusion/exclusion. Patients were excluded based on the following criteria: active systemic (gastrointestinal and non-gastrointestinal) infection; active (solid-organ or hematological) malignancy or treatment with anti-tumor therapies; non-steroidal anti-inflammatory drug use in the last month; hereditary or familial polyposis syndromes; non-IBD forms of colitis including microscopic colitis, ischemic colitis, diversion colitis, or diverticulitis. Eligible patients (see Appendix Table S1) were recruited and consented by the gastroenterologist (signed authorized Participant Information Sheet/Consent Form). For patients with IBD, intestinal biopsies were retrieved endoscopically from: (1) inflamed, (2) non-inflamed, or (3) marginal areas of inflammation. The same exclusion criteria as above were applied to patients without IBD (referred to as non-IBD control patients), with biopsies retrieved endoscopically from non-inflamed segments of the intestine. Boston Scientific Radial Jaw™ biopsy forceps and Olympus EVIS EXERA III endoscopes were routinely utilized for intestinal biopsy collections. Collected biopsies were immediately placed in 10% v/v Neutral Buffered Formalin. A ratio of one part tissue to >10 parts formalin was used. Tissues were incubated in formalin at room temperature for 24–72 h before paraffin-embedding.

## Blinded histopathological scoring of intestinal inflammation

Hematoxylin and eosin-stained slides from formalin-fixed paraffin-embedded biopsies were blindly scored by an anatomical pathologist using the Robarts Histopathology Index (RHI), a validated tool for assessing IBD activity, using previously described methodology (Mosli et al, 2017).

## Cell lines

HT29 cells were originally sourced from the American Type Culture Collection. The $RIPK1^{-/-}$, $RIPK3^{-/-}$, $MLKL^{-/-}$, and $CASP8^{-/-}CASP10^{-/-}MLKL^{-/-}$ HT29 cells have been previously reported (Jacobsen et al, 2022; Petrie et al, 2018; Tanzer et al, 2017).

Mouse dermal fibroblasts were generated in-house from the tails of wild-type C57BL/6J mice and immortalized by SV40 large T antigen as reported previously (Hildebrand et al, 2014). The sex and precise age of these animals were not recorded, although our MDFs are routinely derived from tails from 8-week-old mice. Mouse dermal fibroblast lines were generated in accordance with protocols approved by the Walter and Eliza Hall Institute of Medical Research Animal Ethics Committee. The origin of cell lines was not further verified, although their morphologies and responses to necroptotic stimuli were consistent with their stated origins. Cell lines were monitored via polymerase chain reaction every ~6 months to confirm they were mycoplasma-free.

## Cell culturing

HT29 cells and mouse dermal fibroblasts were maintained in Dulbecco's Modified Eagle Medium (high glucose DMEM; Thermo Fisher Scientific) with 8% v/v fetal calf serum (FCS; Sigma), 2 mM L-glutamine, 50 U/mL penicillin, and 50 U/mL streptomycin (G/P/S). Cells were incubated under humidified 10% $CO_2$ at 37 °C.

## Cell treatment

HT29 cells were treated in DMEM containing 1% v/v FCS and G/P/S. Mouse dermal fibroblasts were treated in DMEM containing 8% v/v FCS and G/P/S. Media for treatment was supplemented with: 100 ng/mL recombinant human TNF-α-Fc (produced in-house as in (Bossen et al, 2006)), 500 nM Smac mimetic/Compound A (provided by Tetralogic Pharmaceuticals as in (Vince et al, 2007)) 5 µM IDN-6556 (provided by Idun Pharmaceuticals). HT29 cells were treated for 7.5 h. Mouse dermal fibroblasts were treated for 2 h.

## Cell lysate preparation

HT29 cells were homogenized in ice-cold RIPA buffer (10 mM Tris-HCl pH 8.0, 1 mM EGTA, 2 mM MgCl2, 0.5% v/v Triton X-100, 0.1% w/v Na deoxycholate, 0.5% w/v SDS, and 90 mM NaCl) supplemented with 1x Protease and Phosphatase Inhibitor Cocktail (Cell Signaling Technology Cat#5872 S) and 100 U/mL Benzonase (Sigma-Aldrich Cat#E1014).

## Cell pellet preparation

Trypsinized cells were centrifuged at $671 \times g$ (2000 rpm) for 3 min at room temperature. The supernatant was discarded, cell pellets resuspended in 10% v/v Neutral Buffered Formalin, incubated for 15 min at room temperature, and centrifuged at $671 \times g$ (2000 rpm) for 3 min at room temperature. Cell pellets were resuspended in 50–70 µL of HistoGel (Epredia Cat#HG-4000-012) pre-warmed to 56 °C and then pipetted onto ice-cold glass coverslips to set. Set pellets were stored in 70% (v/v) ethanol until paraffin-embedding.

## Human tissue lysate preparation

Two intestinal biopsies (each ~1–2 mm³) were pooled, and immediately washed in ice-cold DPBS (Thermo Fisher Scientific Cat#14190144) supplemented with protease inhibitors (Merck

Cat#4693132001) and phosphatase inhibitors (Merck Cat#4906837001), and then homogenized with a stainless steel ball bearing in a Qiagen TissueLyzer II (30 Hz, 1 min) in 0.4 mL of ice-cold RIPA buffer (10 mM Tris-HCl pH 8.0, 1 mM EGTA, 2 mM $MgCl_2$, 0.5% v/v Triton X-100, 0.1% w/v sodium deoxycholate, 0.5% w/v SDS, and 90 mM NaCl) supplemented with 1x Protease and Phosphatase Inhibitor Cocktail (Cell Signaling Technology Cat#5872) and 100 U/mL Benzonase (Sigma-Aldrich Cat#E1014).

## Necrosome quantitation

Necrosome detection was performed with a custom pipeline developed in Fiji (Schindelin et al, 2012). Cells were segmented using CellPose (Stringer et al, 2021) and puncta detected using a difference-of-Gaussian algorithm developed for graphics processing units (Haase et al, 2020). To avoid false-positives, puncta were filtered to be more than twice the signal of the average signal of the cell on which it appears. Number of cells with puncta, along with a number of puncta per cell was recorded. Prior to detecting necrosomes in patient biopsies, the epithelial regions were segmented using FastPathology (v1.1.2) with a deep learning model trained on CD3-stained colon biopsy whole slide images (Pedersen et al, 2021; Pettersen et al, 2021). Briefly, the whole slide images were imported into FastPathology and the default model pipeline was executed using the attribute "patch-level 4". The binary mask was imported into QuPath (v0.4.3) using the "importPyramidal-TIFF.groovy" script provided at https://github.com/andreped/NoCodeSeg (Pedersen et al, 2021).

### The Paper Explained

**Problem**

Necroptosis is a recently described form of programmed cell death proposed to cause various inflammatory conditions in mice and humans. Inflammatory bowel disease (IBD), an incurable chronic gastrointestinal disease with a rising prevalence globally, is one such disease where necroptosis is thought to play a central role and which may benefit from anti-necroptotic therapies. However, a definitive link between necroptosis and disease has remained elusive – in large part because there are no clinically-relevant methodologies for detecting this cell death pathway in tissues.

**Results**

This study developed and rigorously tested methods to immunostain the necroptosis pathway under different conditions in mouse tissues and in tissues from humans with and without IBD. In doing so, we discovered that regulators of necroptosis were differentially expressed in a context-dependent manner in mice and humans. We demonstrate that circulating markers in blood, rather than just tissue, may signify necroptotic cell death in mice. We also find that different modes of cell death arise in IBD, and that our new immunostaining methods can be used to distinguish these varying subroutines of cell death.

**Impact**

This study provides a reproducible manual for the immunodetection of necroptosis that can be applied in both the laboratory and clinical settings. Visualising when and where necroptosis occurs will, in turn, enable identification of patients who may benefit from anti-necroptotic therapies in the future.

## Statistical tests

The number of independent experiments and the employed statistical test for each dataset is stipulated in the respective figure legend. Statistical tests were performed using Prism v.9.5.1 (GraphPad). Statistical analyses were only performed on datasets collated from at least three independent experiments. The number of independent replicates or mice is stipulated by "n", and the number of technical replicates for each dataset is stipulated by "N" in the respective Figure legend. All histopathological scoring was performed in a manner where the pathologist was blinded to the phenotype of the patient.

## Data availability

The RNA-seq datasets produced in this study are available in the following database : Gene Expression Omnibus repository [GSE262762].

The source data of this paper are collected in the following database record: biostudies:S-SCDT-10_1038-S44321-024-00074-6.

## Peer review information

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

## Acknowledgements

We thank James Vince, Jiyi Pang, David Tarlington, Angus Stock, Najoua Lalaoui, Asha Jois, Marcel Doerflinger, Quentin Gouil, Eric Hanssen, Phil Arandjelovic, Imadh Azeez and Bruce Rosengarten for constructive feedback and/or reagents during the preparation of this manuscript. We thank WEHI Monoclonal antibody facility for producing several antibodies used in this study. We thank the WEHI Histology team for high-level support with immunohistochemistry. We thank the WEHI Bioservices team for high-level support for animal experimentation, welfare, and ethics. We thank Kim Newton and Vishva Dixit for sharing the *Ripk3*$^{-/-}$ mice that were used to make the *Casp8*$^{-/-}$*Ripk3*$^{-/-}$ mice, and thank Michelle Kelliher for the *Ripk1*$^{-/-}$ mice that were used to generate the *Casp8*$^{-/-}$*Ripk1*$^{-/-}$*Ripk3*$^{-/-}$ mice. We thank Anaxis Pharma Pty Ltd for providing human RIPK3 expressing mice for our control experiments, and thank Andrew Kueh and Ueli Nachbur for their roles in generating this strain. We are grateful to the Department of Gastroenterology at the Royal Melbourne Hospital for performing endoscopies and to the patients who consented to donating their material to be used in this study. This work was supported by the National Health and Medical Research Council of Australia (grants 1172929 to JMM, 2008652 to EDH, 2002965 to ALS, and the Independent Research Institutes Infrastructure Support Scheme 9000719), by the Kenneth Rainin Foundation (award to JMM, BC, AHA, ALS), and by the Victorian State Government Operational Infrastructure Support scheme. SC is supported by the WEHI Handman PhD scholarship, and AHA by the Avant Foundation, Crohn's and Colitis Australia, and the University of Melbourne Scholarships. JMM and JS received research funding from Anaxis Pharma Pty Ltd, from which the salaries of KMP, AH, AL, and PG were paid.

## Author contributions

**Shene Chiou**: Data curation; Validation; Investigation; Methodology; Writing—original draft. **Aysha H Al-Ani**: Data curation; Validation; Investigation; Methodology; Writing—review and editing. **Yi Pan**: Resources; Data curation; Validation; Investigation; Methodology. **Komal M Patel**: Data curation; Formal analysis; Validation; Investigation; Methodology. **Isabella Y Kong**: Resources; Formal analysis; Visualization. **Lachlan W Whitehead**: Resources; Software; Formal analysis; Visualization; Methodology. **Amanda Light**: Data curation; Formal analysis; Investigation; Methodology. **Samuel N Young**: Resources; Investigation. **Marilou Barrios**: Resources; Data curation; Investigation. **Callum Sargeant**: Resources; Formal analysis; Investigation. **Pradeep Rajasekhar**: Resources; Software; Formal analysis; Methodology. **Leah Zhu**: Methodology. **Anne Hempel**: Data curation; Investigation. **Ann Lin**: Data curation; Investigation. **James A Rickard**: Data curation; Formal analysis; Investigation. **Cathrine Hall**: Resources; Investigation. **Pradnya Gangatirkar**: Data curation; Investigation. **Raymond KH Yip**: Resources; Investigation. **Wayne Cawthorne**: Data curation; Investigation; Methodology. **Annette V Jacobsen**: Resources; Investigation. **Christopher R Horne**: Resources; Data curation; Investigation; Methodology. **Katherine R Martin**: Investigation. **Lisa J Ioannidis**: Resources; Investigation. **Diana S Hansen**: Resources; Investigation. **Jessica Day**: Resources; Investigation. **Ian P Wicks**: Resources; Writing—review and editing.

**Charity Law**: Resources; Formal analysis; Project administration. **Matthew E Ritchie**: Resources; Formal analysis; Project administration. **Rory Bowden**: Resources; Project administration. **Joanne M Hildebrand**: Validation; Investigation; Visualization. **Lorraine A O'Reilly**: Resources; Investigation. **John Silke**: Resources; Supervision. **Lisa Giulino-Roth**: Resources. **Ellen Tsui**: Resources; Data curation; Supervision; Investigation; Methodology. **Kelly L Rogers**: Resources; Supervision. **Edwin D Hawkins**: Resources; Supervision; Investigation; Methodology; Project administration. **Britt Christensen**: Resources; Supervision; Project administration; Writing—review and editing. **James M Murphy**: Conceptualization; Resources; Supervision; Funding acquisition; Investigation; Methodology; Writing—original draft; Project administration; Writing—review and editing. **André L Samson**: Conceptualization; Resources; Data curation; Software; Formal analysis; Supervision; Funding acquisition; Validation; Investigation; Visualization; Methodology; Writing—original draft; Project administration; Writing—review and editing.

Source data underlying figure panels in this paper may have individual authorship assigned. Where available, figure panel/source data authorship is listed in the following database record: biostudies:S-SCDT-10_1038-S44321-024-00074-6.

## Disclosure and competing interests statement

KMP, SNY, AH, AL, PG, CRH, JD, IPW, JMH, JS, JMM, and ALS have contributed to the development of necroptosis pathway inhibitors in collaboration with Anaxis Pharma Pty Ltd. YP is a co-holder of patent for material arising from this study. All other authors have no additional financial interests.

# Expanded View Figures

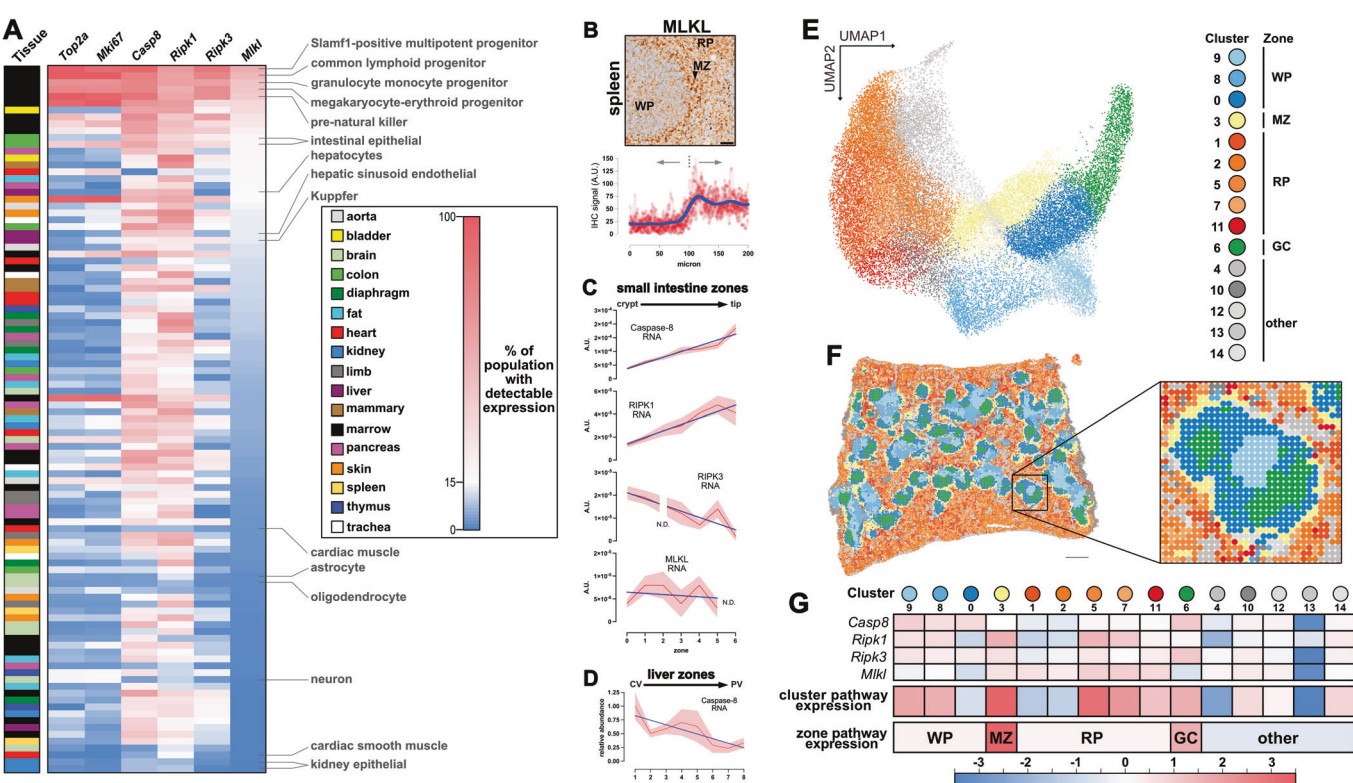

**Figure EV1. Constitutive co-expression of necroptotic effectors is confined to fast-cycling cells within progenitors, immune and barrier populations.**

(**A**) Heatmap of cell ontologies from the *Tabula Muris* dataset (Tabula Muris et al, 2018). Left-most column depicts the tissue origin of each cell ontology. Other columns indicate the percent of cells within each ontology that expressed *Top2a, Mki67, Casp8, Ripk1, Ripk3*, or *Mlkl*. Legend shows the color-to-tissue and the color-to-frequency scales. Cell ontologies of interest are annotated. (**B**) Micrograph of MLKL immunosignals from the wild-type mouse spleen. The white pulp (WP), marginal zone (MZ), and red pulp (RP) are annotated. Scale bar is 50 μm. Scatterplot shows relative expression levels of MLKL along the white pulp-to-red pulp axis. Red datapoints show immunosignal intensities and the overlaid dark blue line indicates the LOWESS best-fit along *N* = 20 axes from *n* = 1 mouse. Dashed line indicates the boundary between splenic white pulp and marginal zone. Data were representative of *n* > 3 mice. (**C, D**) Spatial transcriptomic data from (Moor et al, 2018)) and (Ben-Moshe et al, 2019) showing the relative expression levels (arbitrary units; A.U.) of Caspase-8, RIPK1, RIPK3, or MLKL along the ileal crypt-to-villus axis (**C**) or the hepatic central vein-to-portal vein axis (**D**). (**E–G**) Spatial transcriptomic data on mouse spleen 12 days after *Plasmodium berghei*-infection. Panel (**E**) shows a uniform manifold approximation and projection (UMAP) of cell populations distinguished by unsupervised leiden clustering. Legend shows the color assigned to each population. Panel (**F**) shows the location of each cell cluster. Scale bar is 500 μm. Panel (**G**) shows the normalized expression for each gene product. Expression values for *Casp8, Ripk1, Ripk3*, and *Mlkl* were summated to provide an index of "cluster pathway expression", which was averaged to provide an index of "zone pathway expression". Data were from *n* = 1 mouse.

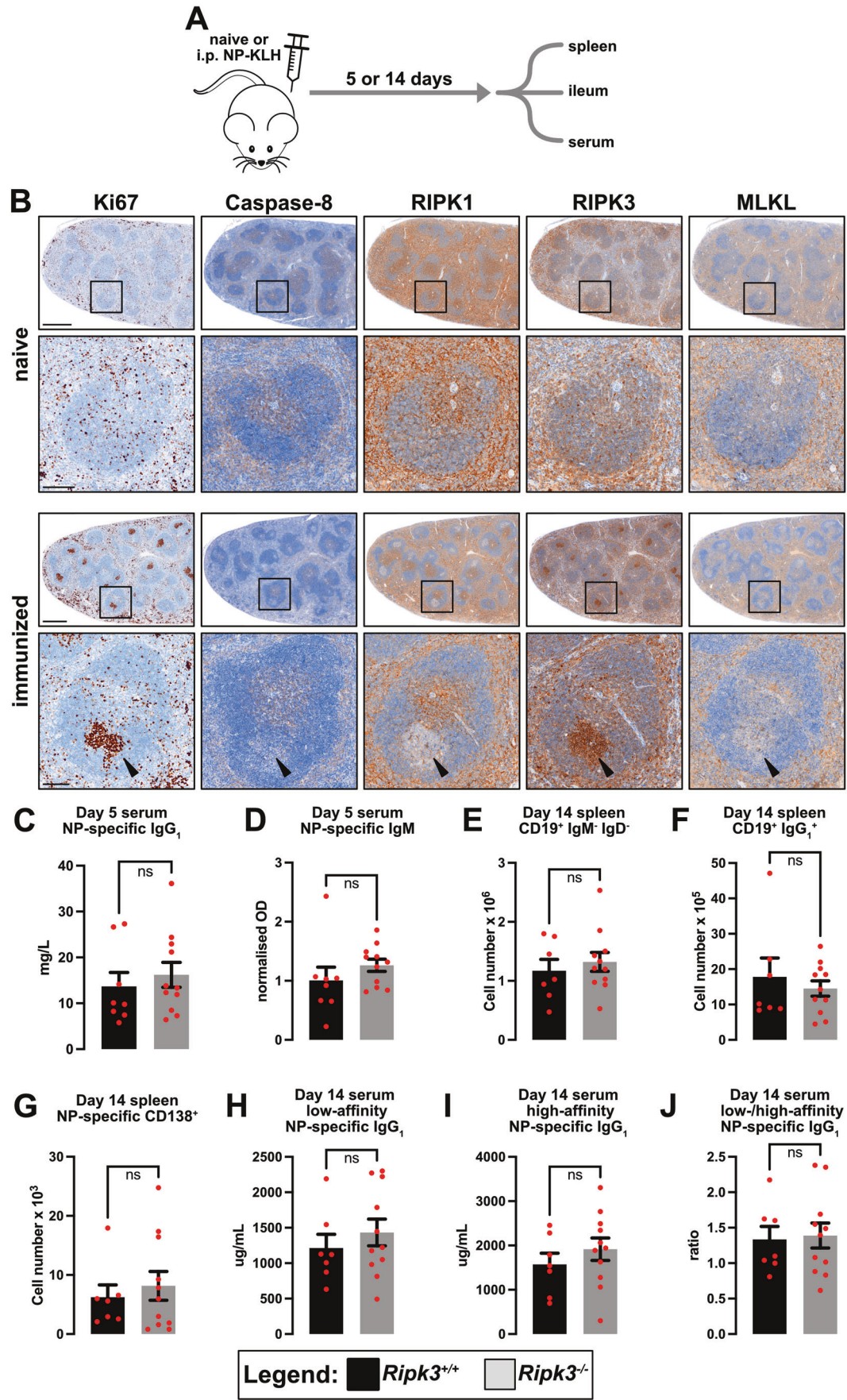

◀   **Figure EV2.   RIPK3 is uniquely upregulated in splenic germinal centers.**

(A) Experimental design. (B) Ki67, Caspase-8, RIPK1, RIPK3, and MLKL immunosignals from adjacent sections of the naïve or NP-KLH-immunized mouse spleen. Arrowheads show a Ki67$^+$ germinal center that co-stains for RIPK3, but not other members of the pathway. Representative of $n > 3$ mice per group. Scale bars in lower magnification micrographs are 500 µm. Scale bars in insets are 100 µm. Data were representative of $n > 3$ mice per group. (C–J) $Ripk3^{-/-}$ or $Ripk3^{+/+}$ mice were immunized with NP-KLH and circulating NP-specific IgG$_1$ (C), circulating NP-specific IgM (D), splenic mature B cells (E, F), splenic NP-specific plasma cells (G), circulating low affinity NP-specific IgG$_1$ antibody (H), circulating high affinity NP-specific IgG$_1$ (I), and the ratio between circulating low-and-high affinity NP-specific antibody (J) were measured at the indicated day after immunization. Bars on graphs in (C–J) represent mean ± SEM. Each datapoint represents one mouse. ns non-significant by two-sided $t$-test with Welch's correction.

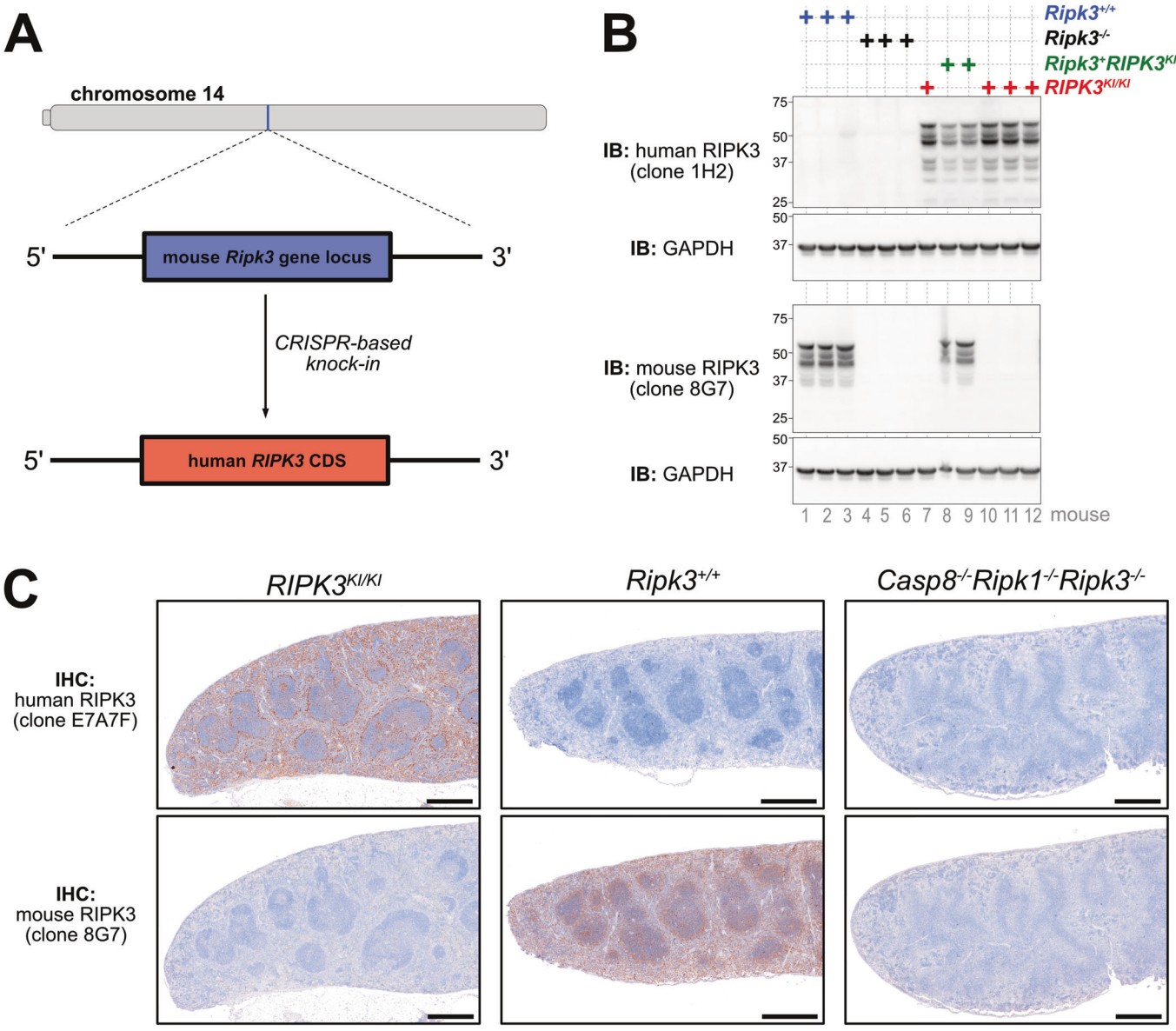

**Figure EV3. Assessing the specificity of immunostaining using wild-type, knockout and human RIPK3 knock-in mice.**

(A) Approach used to insert the human RIPK3 coding sequence (CDS) into the mouse *Ripk3* locus. (B) Immunoblot of spleen homogenates from *Ripk3+/+*, *Ripk3-/-*, human RIPK3 (*RIPK3KI/KI*) or hemizygous human RIPK3 (*Ripk3+RIPK3KI*) mice. GAPDH immunoblots are shown as loading controls. Each lane represents a different mouse. (C) Immunosignals produced by the anti-human RIPK3 (clone 37A7F) or anti-mouse RIPK3 (clone 8G7) antibodies on spleen sections from *RIPK3KI/KI*, *Ripk3+/+* or *Casp8-/-Ripk1-/-Ripk3-/-*. Data were representative of n ≥ 3 for each target and tissue. Scale bars are 500 μm.

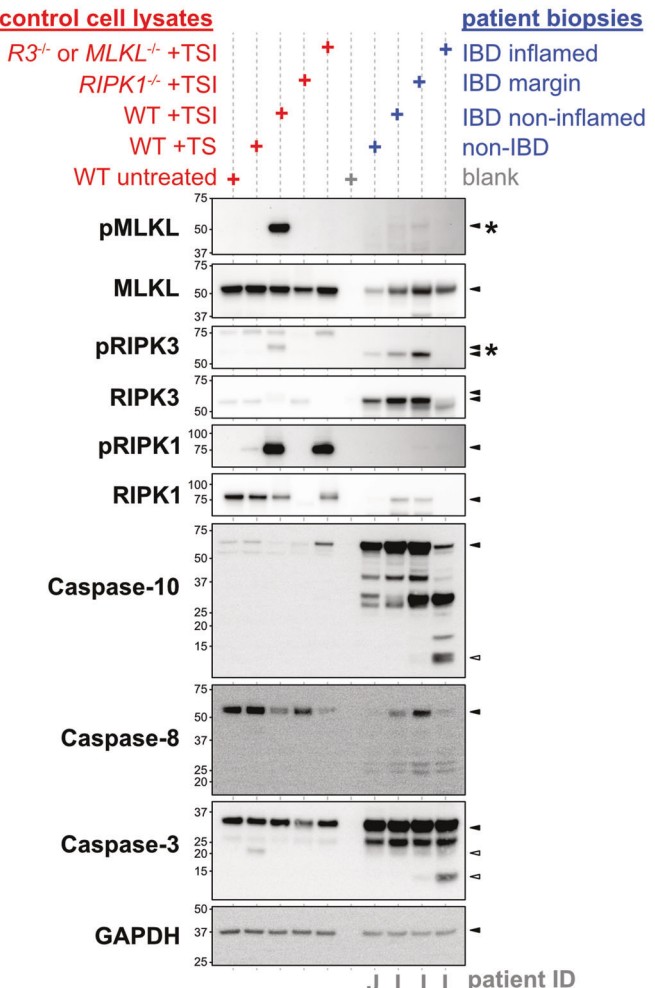

**Figure EV4. Another instance of elevated intestinal apoptosis in a patient with ulcerative colitis.**

Immunoblot of lysates from HT29 cells (red annotations) and intestinal biopsies from patients A-H (blue annotations). The fifth lane of each gel contained lysates from TSI-treated *RIPK3−/−* or TSI-treated *MLKL−/−* cells (see source data for details). Patient J was a non-IBD control. Patient I had ulcerative colitis (UC). The endoscopic grading of the biopsy site as "non-inflamed", "marginally inflamed", or "inflamed" is stipulated. Closed arrowheads indicate full-length form of proteins. Asterisks indicate active, phosphorylated forms of RIPK3 (pRIPK3) and MLKL (pMLKL). Open arrowheads indicate active, cleaved forms of Caspase-10 and Caspase-3. GAPDH was used as a loading control.

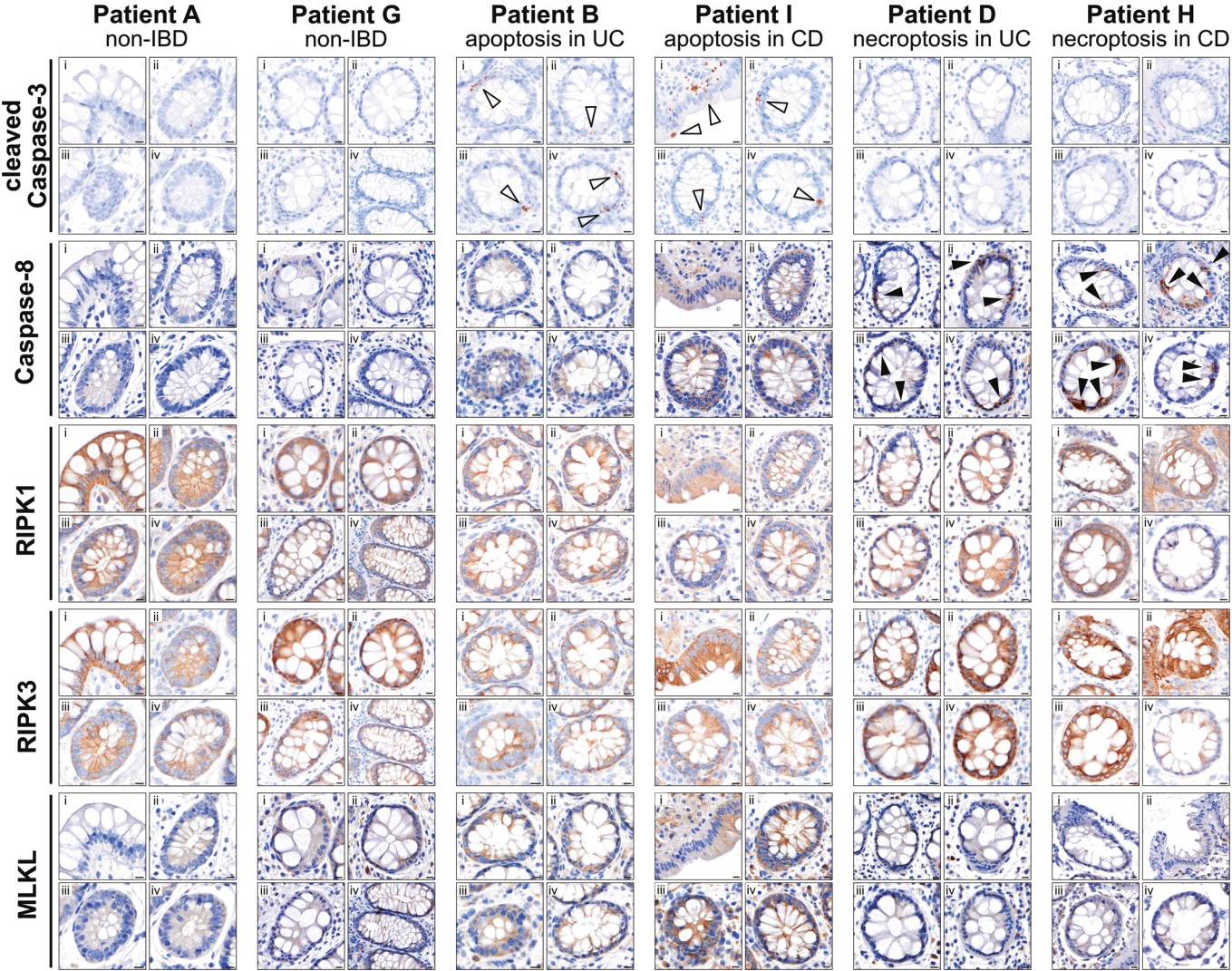

**Figure EV5. Atlas of necroptotic pathway expression in human intestinal crypts.**

Immunohistochemistry for cleaved Caspase-3, Caspase-8 (clone B.925.8), RIPK1, RIPK3, and MLKL (clone EPR171514) on intestinal biopsies from the stipulated patients. Four representative micrographs per biopsy are shown (i–v). Open arrowheads indicate instances of epithelial apoptosis. Closed arrowheads indicate instances of epithelial Caspase-8 clustering. Scale bars are 10 μm. The location for each micrograph within the biopsy is indicated in Appendix Fig. S5.

