## [Peer Review File · EMBO Molecular Medicine]

An immunohistochemical atlas of necroptotic pathway expression

Shene Chiou, Aysha Al-Ani, Yi Pan, Komal Patel, Isabella Kong, Lachlan Whitehead, Amanda Light, Samuel Young, Marilou Barrios, Callum Sargeant, Pradeep Rajasekhar, Leah Zhu, Anne Hempel, Ann Lin, James Rickard, Cathrine Hall, Pradnya Gangatirkar, Raymond Yip, Wayne Cawthorne, Annette Jacobsen, Christopher Horne, Katherine Martin, Lisa Ioannidis, Diana Hansen, Jessica Day, Ian Wicks, Charity Law, Matthew Ritchie, Rory Bowden, Joanne Hildebrand, Lorraine O'Reilly, John Silke, Lisa Giulino-Roth, Ellen Tsui, Kelly Rogers, Edwin Hawkins, Britt Christensen, James Murphy, and Andre Samson

Corresponding authors: Andre Samson (samson.a@wehi.edu.au) , James Murphy (jamesm@wehi.edu.au)

Review Timeline:

Submission Date:	8th Nov 23
Editorial Decision:	29th Nov 23
Revision Received:	29th Mar 24
Editorial Decision:	16th Apr 24
Revision Received:	19th Apr 24
Accepted:	22nd Apr 24

Editor: Zeljko Durdevic

Transaction Report:

29th Nov 2023

Dear Dr. Samson,

Thank you for the submission of your manuscript to EMBO Molecular Medicine. We have now received feedback from the three reviewers who agreed to evaluate your manuscript. All three referees recognize potential interest of the study but also raise important criticism that should be addressed in a major revision. Spatial transcriptomic experiment suggested by the referee #3 is welcomed but not required for further consideration of the manuscript. If you would like to discuss further the points raised by the referees, I am available to do so via email or video. Let me know if you are interested in this option.

We would welcome the submission of a revised version within three months for further consideration. Please let us know if you require longer to complete the revision.

I look forward to receiving your revised manuscript.

Yours sincerely,

Zeljko Durdevic

We require:

- 1) A .docx formatted version of the manuscript text (including legends for main figures, EV figures and tables). Please make sure that the changes are highlighted to be clearly visible.
- 2) Individual production quality figure files as .eps, .tif, .jpg (one file per figure). For guidance, download the 'Figure Guide PDF': (<https://www.embopress.org/page/journal/17574684/authorguide#figureformat>).
- 3) A .docx formatted letter INCLUDING the reviewers' reports and your detailed point-by-point responses to their comments. As part of the EMBO Press transparent editorial process, the point-by-point response is part of the Review Process File (RPF), which will be published alongside your paper.
- 4) A complete author checklist, which you can download from our author guidelines (<https://www.embopress.org/page/journal/17574684/authorguide#submissionofrevisions>). Please insert information in the checklist that is also reflected in the manuscript. The completed author checklist will also be part of the RPF.
- 5) Please note that all corresponding authors are required to supply an ORCID ID for their name upon submission of a revised manuscript.
- 6) It is mandatory to include a 'Data Availability' section after the Materials and Methods. Before submitting your revision, primary

datasets produced in this study need to be deposited in an appropriate public database, and the accession numbers and database listed under 'Data Availability'. Please remember to provide a reviewer password if the datasets are not yet public (see <https://www.embopress.org/page/journal/17574684/authorguide#dataavailability>).

13) Author contributions: You will be asked to provide CRediT (Contributor Role Taxonomy) terms in the submission system. These replace a narrative author contribution section in the manuscript.

14) A Conflict of Interest statement should be provided in the main text.

Please also suggest a striking image or visual abstract to illustrate your article as a PNG file 550 px wide x 300-800 px high.

**** Reviewer's comments ****

Referee #1 (Remarks for Author):

The paper established, optimized and validated a set of IHC staining protocols to detect critical components of the necrosome in fixed mouse and human tissue. This set of protocols were applied to examine the activation of necroptosis in a variety of tissues and conditions in mice and human, including IBD patients. Using their optimized protocols, the authors make the remarkable finding of heterogeneity in the mode of cell death within intestinal tissue biopsies from both CD and UC, where apoptosis prevailed in some patients and necroptosis prevailed in others. The authors also show surprisingly restricted expression of the necroptosis orchestrator molecule RIPK3 (and MLKL) in mouse tissues under homeostasis and its preferential expression in cycling cells and also within plasma cells following immunization.

Strengths:

1. Systematic optimization and validation methodology
2. Focus on a challenging problem many investigators face in being able to reliably detect and identify necroptosis in tissue
3. Extensive characterization of necroptotic pathway components over different tissue, species, and conditions cross-validated with other techniques and published databases
4. Revealed pattern of in situ necroptosis in human IBD
5. Well written text and thoroughly disclosed methodology

Specific concerns:

1. Has caspase 8 been shown to be part of the necrosome when necroptosis is underway? The authors should include these references.
2. The authors speculate that the punctate staining in Fig. 4D is due to chylomicrons. This speculation is confusing and needs an explanation. Is punctate staining in general attributed to chylomicrons and does that imply that the same punctate staining in human tissue is indicative of the same? Does this reflect an association with dysbiosis which is a hallmark of IBD?
3. Fig 5A: The authors note co-localization of caspase-8, RIPK1, and MLKL in TSI cells and state that all three molecules are held at necrosomes. However, to make this conclusion, the cells need to be stained simultaneously for the three markers and areas of colocalization identified. The presence of all three markers within the same group of treated cells does not mean that they can all colocalize in the same cell.
4. Fig 6D: Intestinal tissue (colon/rectum) from patient D was characterized as 'necroptotic' rather than 'apoptotic'. The authors use caspase-8 staining pattern here to suggest that cytoplasmic caspase-8 clusters may represent bone fide necrosomes (lines 326-327). This is confusing in light of the data in Fig. 6C showing pMLKL but not cleaved caspase-8. The authors state that there were no obvious changes to the established markers of RIPK1 and MLKL but those data were not shown and should be included to illustrate the point. It is also important to look at a second 'necroptosis prone' patient, patient H for example.

Concurrently the same should be done on intestinal tissue from 'apoptosis' prone patients B and F. If caspase-8 clusters are also observed in intestinal tissue from 'apoptosis' prone patients, then the suggestion cannot be made that caspase-8 can be used as a bone fide necrosome marker. The suggestion is therefore currently unwarranted and may even lead to confusion in the field.

5. When the authors conducted optimization of detecting the human necrosome, they used fixed, paraffin embedded cell pellets from HT-29 cell line (line 255-258), and the protocol which worked on HT29 cells was used on human tissue sections. Can the authors elaborate on their rationale in the text of the manuscript to provide justification for the extrapolation from cell lines to tissue sections, especially considering that the fixation conditions appear to be different between the HT-29 cell pellets and the human tissues.

6. The authors show that injection of TNF into mice changes the intensity and pattern of expression of RIPK3, but they fall short of examining the same changes by IHC within IBD non-inflamed, margin and IBD inflamed biopsy tissues. This is especially when they had invested considerable effort in optimizing and validating the detection methods, with the specific goal of looking in IBD tissues. Immunoblots do not reveal the tissue spatial distribution. They have the tissues, and this should have been done.

Minor:

1. Abbreviations text and figures should be explained, for example TS and TSI, endo and Kpf, RHI and so on.
2. Mark the MW sizes in each group of immunoblots 'necroptosis in UC', 'apoptosis in CD', and 'necroptosis in CD' and the tick marks are different for these immunoblots compared to the tick marks labeled for the leftmost group 'apoptosis in UC'.
3. Fig. 6D should be labeled to make it clear that the staining pattern shown is that of caspase-8.

Referee #2 (Remarks for Author):

Necroptosis is a form of lytic cell death that is important for host defence, however, its dysregulation is also associated with an increasing list of human disease. However, as the authors pointed out, many of these studies are contradictory and the field has yet to reach a consensus on the contribution of necroptosis to various diseases. This is in part, due to the lack of specific reagents to assess necroptotic pathway in mice and human. This manuscript addresses this issue and is an extremely valuable tool for the cell death community. I believe this study, together with their previous toolbox paper, will help the field significantly. I have some suggestions that I hope will be useful to improve the manuscript and the cell death community.

1. Can the authors also include the concentration in which the antibodies were used for both immunoblot and IHC? In my experience, some antibodies such as the CST cleaved caspase-8 works much better when blocked with BSA instead of milk; is this also the case for these antibodies? If so, the authors should include the optimal blocking reagent for each antibody in the methods.
2. I am curious (and frankly a little sceptical) about the multiple necrosome puncta observed in HT-29 cells (Fig 5C) since inflammatory foci or innate immune signalling hubs typically collapse into one single puncta. This is observed for example during RIG-I/MAVS, TLR4 and inflammasome-ASC. Are the authors confident that these are not artefacts?
3. I appreciate that patient samples are previous and limited. However, I find it hard to draw meaningful conclusion for Figure 6 the authors are basically comparing one healthy control to one IBD patient per group (please correct me if I am wrong). This will be much more convincing with a few more samples.
4. In Fig 6D, using IHC, the authors found caspase-8 puncta in inflamed IBD patients and propose this as a potential unbiased method to detect the necrosome. I disagree with this definition since caspase-8 puncta can also represent the ripoptosome/Complex II that drives apoptosis. Since the human MLKL antibodies seem to work in HT-29 cells (Fig. 5A), and yet only caspase-8 but MLKL puncta are not detected in human samples, I would instead conclude that inflamed IBD tissues are associated with apoptosis, not necroptosis (at least based on 1 patient sample).

Referee #3 (Remarks for Author):

This manuscript by Chiou et al describes a standardized process for immunohistochemical assessment of necroptosis mediators in murine and human tissue sections. The clinical implications of reproducibly characterizing cell death activity is an important goal. The authors optimize a number of antibodies and validate these antibodies. They use these reagents to examine the expression of several core cell death mediators, including caspase 8, RIP1, and RIP3. They localize differential expression of some of these markers to distinct cell subtypes in murine intestinal tissues, and note selective changes in response to TNF, antibiotics and to immunogens.

The general idea of optimizing immunohistochemical quantification of these important proteins is well reasoned, given the importance of these proteins to cellular survival. It is a little disappointing that MLKL could not be studied alongside RIP3, given

the intriguing paradigm of RIP3 dependent MLKL necroptosis processes versus non-necroptosis RIP3 dependent processes. Standardization of these assays for use upon formalin fixed tissues could significantly enhance our understanding of the potential roles of necroptosis in various diseases.

Issues that should be addressed include:

1. Expression levels of RIP3 were originally identified as conferring susceptibility to necroptosis by Xiaodang Wang's lab (He et al, Cell 2009). Chiou et al found selective expression of RIP3 in subsets of intestinal epithelial cells and splenocytes in unperturbed mice, reinforcing the idea that this protein may be particularly important in regulating basal homeostasis of these cells.

As the authors associate increased RIP3 expression with robustly proliferating cells, and as necroptosis is experimentally increased when apoptosis mediators are inhibited (e.g., ZVAD blockade of caspases). Do the authors think that increased proliferation and caspase inhibition enhance necroptosis via similar/overlapping or distinct mechanisms. In the latter scenario, additive or synergistic necroptosis might be observed in certain scenarios. This issue might be inferred from histochemical data, but could also be tested experimentally.

2. The complexity and activity of cell death pathways might be enhanced by broadening the panel to include other well characterized regulators of this process, e.g., phosphorylation of RIP3 is thought to reflect RIP3 activation during necroptosis signaling, and antibodies specific for phosphorylated RIP3 have been described. Could the authors test these antibodies to potentially enhance the detection of active necroptosis?

3. Along similar lines as point 2 above, it would be interesting to know whether phosphorylation of RIPK1 is increased. We understand that caspase 8 cleavage would be difficult to localize by immunoblotting, but there are antibodies specific for caspase 8 cleavage products.

4. The notion that RIP3 expression is an acute phase reactant is interesting, and might align with the cell biological notion of inflammatory cell death. Acute phase reactants have clinically been considered in the context of inflammatory serum markers (i.e., secreted proteins). Do the authors think that RIP3 might be detectable in serum, perhaps released from dying cells that are not quietly phagocytosed by neighboring cells?

5. Systemic TNF injection induced RIPK3 expression in IECs, selected vascular beds and liver, while caspase 8 and RIPK1 expression were not (Fig. 3). This observation raises questions about the regulation of these proteins. Can the authors gain some insights into the mRNA expression levels of these genes via some type of spatial transcriptomic assay?

6. The shift of RIPK1 and RIPK3 expression within IEC subsets is interesting, and begs the question of whether antibiotic depletion alters IEC transit time. Could this be addressed via BrdU pulse expt?

7. Heterogeneity in cell death signaling proteins in human IBD patients are very interesting. Can the authors provide some clinical information regarding the patients? Did the patients with increased signs of necroptosis display greater signs of intestinal or systemic inflammation?

We thank the reviewers for their constructive feedback and appraisal of our work. It was pleasing that the reviewers recognised our efforts to automate, optimise and cross-validate a panel of immunohistochemical stains for the necroptotic pathway. It was also pleasing that the reviewers thought this approach would be of value to the cell death community and recognised its clinical implications. In response to the reviewer queries (in blue italicised text), we have included our replies (in black text) below. We have highlighted changes in the main text in green shading for additions and red shading for deleted text. Furthermore, in response to the reviewers' queries, we have added 5 entirely new figures (Fig. **EV3, 7, 8, 9, 10**), and added new panels to 6 existing display items (Fig. **3K-L, 5A, 6B, EV4C-J, EV5** and Table 1).

Referee #1 (Remarks for Author):

The paper established, optimized and validated a set of IHC staining protocols to detect critical components of the necrosome in fixed mouse and human tissue. This set of protocols were applied to examine the activation of necroptosis in a variety of tissues and conditions in mice and human, including IBD patients. Using their optimized protocols, the authors make the remarkable finding of heterogeneity in the mode of cell death within intestinal tissue biopsies from both CD and UC, where apoptosis prevailed in some patients and necroptosis prevailed in others. The authors also show surprisingly restricted expression of the necroptosis orchestrator molecule RIPK3 (and MLKL) in mouse tissues under homeostasis and its preferential expression in cycling cells and also within plasma cells following immunization.

Strengths:

- 1. Systematic optimization and validation methodology*
- 2. Focus on a challenging problem many investigators face in being able to reliably detect and identify necroptosis in tissue*
- 3. Extensive characterization of necroptotic pathway components over different tissue, species, and conditions cross-validated with other techniques and published databases*
- 4. Revealed pattern of in situ necroptosis in human IBD*
- 5. Well written text and thoroughly disclosed methodology*

Specific concerns:

1. Has caspase 8 been shown to be part of the necrosome when necroptosis is underway? The authors should include these references. Caspase-8-deficient models show that necrosomes can per se form in the absence of Caspase-8. But, there are many studies showing that Caspase-8, when expressed, is recruited to necrosomes. For example, de Almagro MC et al., 2017 *CDD* used Caspase-8 immunoprecipitation to fractionate necrosomes from cells undergoing TNF-induced necroptosis. Other studies that use co-immunoprecipitation to show that Caspase-8 is recruited to the necrosome include: He S. et al., 2009 *Cell*, Li X. et al., 2021 *Protein Cell* and Sun L. et al., 2012 *Cell*. To highlight this point, the following text and citations have been added to line 293: "orthogonal approaches show that Caspase-8, RIPK1, RIPK3 and MLKL are recruited to necrosomes during TNF-induced necroptosis (de Almagro et al, 2017; He et al., 2009; Li et al, 2021; Sun et al., 2012)."

2. The authors speculate that the punctate staining in Fig. 4D is due to chylomicrons. This speculation is confusing and needs an explanation. We initially hypothesised that RIPK1⁺ RIPK3⁺ clusters form in antibiotic-treated mice because of PAMP-containing chylomicrons; which would locate to enterocytes at villi tips. However, we understand this hypothesis may be overly speculative, and so we have deleted it from the manuscript. Instead, we have performed new experiments to determine whether the RIPK1⁺ RIPK3⁺ clusters in antibiotic-treated mice function as necrosomes. In addition, we have provided a more general hypothesis about why these RIPK1⁺ RIPK3⁺ clusters preferentially arise in enterocytes at villi tips. These additions can be found on line 234: "These RIPK1⁺ RIPK3⁺ Caspase-8⁻ clusters are unlikely to be necrosomes, as no

corresponding phospho-activation of RIPK1 or MLKL was observed (Fig. EV3). Instead, these clusters may be due to a microbe-related function, such as lipopolysaccharide handling, that is preferentially performed by enterocytes at villi tips (Berkova et al, 2023; Ge et al, 2000).”

Is punctate staining in general attributed to chylomicrons and does that imply that the same punctate staining in human tissue is indicative of the same? Does this reflect an association with dysbiosis which is a hallmark of IBD? To avoid excess speculation, we have not commented about whether the RIPK1⁺ RIPK3⁺ Caspase-8⁻ clusters that form in antibiotic-treated mice are relevant to dysbiosis-associated IBD. However, as shown in the new Fig. EV9, no overt clustering of epithelial RIPK1 or RIPK3 was observed in IBD patients. It is difficult to draw further conclusions here as the state of microbiome in these IBD patients is currently the subject of ongoing investigation.

3. Fig 5A: The authors note co-localization of caspase-8, RIPK1, and MLKL in TSI cells and state that all three molecules are held at necrosomes. However, to make this conclusion, the cells need to be stained simultaneously for the three markers and areas of colocalization identified. The presence of all three markers within the same group of treated cells does not mean that they can all colocalize in the same cell. We previously showed that RIPK1 and MLKL colocalise at necrosomes in TSI-treated HT29 cells (Samson et al., 2020 Nat Comms). We nonetheless attempted to show that Caspase-8, RIPK1 and MLKL are recruited to the same necrosomes by combining automated IHC with Opal multiplexing. These experiments confirmed that RIPK1 and MLKL colocalise at necrosomes (**Response Figure 1**). However, for unknown reasons, our Caspase-8 staining protocols were not compatible with Opal multiplexing. Because of this, we purposefully avoid the term “co-localise” or similar terms in the manuscript.

Response Figure 1. Representative micrographs of wild-type (WT) or *RIPK1*^{-/-} or *CASP8*^{-/-}*CASP10*^{-/-}*MLKL*^{-/-} HT29 cells that were untreated or TSI-treated for 7.5 hours. Formalin-fixed paraffin-embedded pellets were then made from cells, sections cut and co-stained for DAPI, MLKL (clone EPR17514), Caspase-8 (clone B.925.9) and RIPK1 (clone D94C12) using the same protocols as in File EV1 except that Opal multiplex detection reagents were used instead of standard DAB detection reagents. Sections were imaged on a Vectra Polaris Imaging System (Akoya Biosciences) using default Opal filters. Insets in ‘low mag’ micrographs relate to ‘medium mag’ micrographs. Insets in ‘medium mag’ micrographs relate to ‘high mag’ micrographs. Arrowheads indicate MLKL⁺ clusters that co-stain for RIPK1⁺. Please note, the Caspase-8 signal is non-specific as it was detected in both WT and *CASP8*^{-/-}*CASP10*^{-/-}*MLKL*^{-/-} HT29 cells.

4. Fig 6D: Intestinal tissue (colon/rectum) from patient D was characterized as 'necroptotic' rather than 'apoptotic'. The authors use caspase-8 staining pattern here to suggest that cytoplasmic caspase-8 clusters may represent bone fide necrosomes (lines 326-327). This is confusing in light of the data in Fig. 6C showing pMLKL but not cleaved caspase-8. We apologise for any lack of

clarity with our explanations. We do not believe these observations are contradictory. For example, clusters of Caspase-8 arise in TSI-treated HT29 cells, when Caspase-8 is almost exclusively in its full-length form (compare Fig. 5 and Fig. EV5). Similar events could explain why immunoblot data shows that pMLKL and full-length Caspase-8 coincide with immunohistochemical evidence of Caspase-8+ clusters in patient D.

The authors state that there were no obvious changes to the established markers of RIPK1 and MLKL but those data were not shown and should be included to illustrate the point. We thank the reviewer for raising this important point. We fully agree and these data have now been included in Fig. EV9-10.

It is also important to look at a second 'necroptosis prone' patient, patient H for example. Concurrently the same should be done on intestinal tissue from 'apoptosis' prone patients B and F. If caspase-8 clusters are also observed in intestinal tissue from 'apoptosis' prone patients, then the suggestion cannot be made that caspase-8 can be used as a bone fide necrosome marker. The suggestion is therefore currently unwarranted and may even lead to confusion in the field. We agree with the reviewer that this is an important point. Our initial goal was to provide proof of concept in this work, although we absolutely agree that we needed to generate additional data to be able to draw this conclusion with certainty. In light of the reviewer's comments, we have now applied the full panel of immunohistochemical stains to patients A-E and G-J. Overt clusters of epithelial Caspase-8 were also observed in necroptosis-prone patient, patient H (Fig. EV9-10), in addition to the originally presented data derived from biopsies from the necroptosis-prone patient D. Conversely, no pronounced clusters of epithelial Caspase-8 were observed in the apoptosis-prone IBD patients, patients B and I (Fig. EV9-10). These observations align with our cell culture studies where Caspase-8 clusters are observed in necroptotic HT29 cells, but not in apoptotic HT29 cells (Fig. 5). Altogether, these two lines of evidence support the idea that pronounced clusters of Caspase-8 are a marker of epithelial necroptosis.

Note, patient F had severe colitis at the time of endoscopy, and so we only collected biopsies for histopathology scoring and for immunoblotting. No matched biopsies were collected from patient F for immunohistochemistry. Accordingly, we added patient I (another Crohn's disease patient with signs of increased apoptosis) to this study. We updated the main text to describe these additional data in lines 347.

5. When the authors conducted optimization of detecting the human necrosome, they used fixed, paraffin embedded cell pellets from HT-29 cell line (line 255-258), and the protocol which worked on HT29 cells was used on human tissue sections. Can the authors elaborate on their rationale in the text of the manuscript to provide justification for the extrapolation from cell lines to tissue sections, especially considering that the fixation conditions appear to be different between the HT-29 cell pellets and the human tissues. We apologise for any confusion. The cell pellets and tissues used in this paper are formalin-fixed paraffin-embedded (FFPE), and thus processed in the same way. Nevertheless, the reviewer is correct that our study assumes that immunohistochemistry protocols optimised on FFPE cell pellets retain their specificity when applied to FFPE tissues. New data that addresses this issue has been added to the manuscript and outlined on line 310: "To test this assumption, we used two antibodies - one specific for mouse RIPK3 and one specific for human RIPK3 - on spleens from *Ripk3*^{-/-} mice, *Ripk3*^{+/+} mice or knock-in mice expressing human RIPK3 (see *Methods*). As shown in Fig. EV7, immunoblotting and immunohistochemistry with the respective antibodies accurately discriminated between the expression of mouse RIPK3 or human RIPK3 in spleen. These data suggest that immunohistochemistry protocols optimised on cell pellets can also be used to specifically stain tissues."

6. The authors show that injection of TNF into mice changes the intensity and pattern of expression of RIPK3, but they fall short of examining the same changes by IHC within IBD non-inflamed, margin and IBD inflamed biopsy tissues. This is especially when they had invested

considerable effort in optimizing and validating the detection methods, with the specific goal of looking in IBD tissues. Immunoblots do not reveal the tissue spatial distribution. They have the tissues, and this should have been done.

Like the reviewer, we are also interested in determining whether RIPK3 is regulated as an acute phase reactant in humans, as it is in mice. Our data shows that the expression of full-length RIPK3 is increased in some IBD patients (patients D, H and I relative to non-IBD controls; Fig. **6C** and **EV8**), whereas RIPK3 expression is decreased in other IBD patients (patients B and F relative to non-IBD controls; Fig. **6C**). Such patient-to-patient variability is expected, given that human biopsy studies are inherently more variable than our tightly controlled mouse studies where TNF is given to previously unchallenged co-housed littermates (Fig. **3**). Accordingly, we believe that RIPK3 expression in a higher number of patients needs to be analysed before firm conclusions can be drawn on this topic. To highlight the need for future clinical cohort studies, we have added the following statement to line 483: “and raises questions about whether RIPK3 expression is also regulated similarly in humans.” It is also worth noting that only with the gift of a human *RIPK3* knockin mouse were we able to validate a new human RIPK3 antibody as a suitable reagent for staining human tissue sections while this manuscript was under review. This is a major advance; the reviewer is likely aware that no suitable reagents to stain for human RIPK3 have been available until this discovery. We expect reporting of our validation of this reagent will prove important for the advancement of this area of research, just as it positions us for future detailed studies of biopsies in our patient cohort.

Minor:

1. Abbreviations text and figures should be explained, for example TS and TSI, endo and Kpf, RHI and so on. We apologise for these omissions. Definitions for TNF (T), Smac mimetic (S) and IDN-6556 (I) have now been added to the main body of text on line 289. The abbreviations Endo, Kpf and RHI, as well as all other abbreviations are stipulated in the respective figure legends as well as in the *Abbreviations* section.

2. Mark the MW sizes in each group of immunoblots 'necroptosis in UC', 'apoptosis in CD', and 'necroptosis in CD' and the tick marks are different for these immunoblots compared to the tick marks labeled for the leftmost group 'apoptosis in UC'. While we take the reviewer's point, we would prefer not to add more text alongside every blot panel to avoid crowding Fig. **6C**. Instead, we rely on the horizontal alignment of the markers between the respective panels in Fig. **6C**. Of course, if the reviewer and editors disagree with our approach, we will gladly fully annotate every immunoblot panel in Fig. **6C**. We also note that uncropped images of all immunoblots, together with fully annotated molecular weight markings, are now available as source material.

3. Fig. 6D should be labeled to make it clear that the staining pattern shown is that of caspase-8. We thank the reviewer for this suggestion. The micrographs in Fig. **6D** have now been annotated to indicate that the immunosignals are for Caspase-8.

Referee #2 (Remarks for Author):

Necroptosis is a form of lytic cell death that is important for host defence, however, its dysregulation is also associated with an increasing list of human disease. However, as the authors pointed out, many of these studies are contradictory and the field has yet to reach a consensus on the contribution of necroptosis to various diseases. This is in part, due to the lack of specific reagents to assess necroptotic pathway in mice and human. This manuscript addresses this issue and is an extremely valuable tool for the cell death community. I believe this study, together with their previous toolbox paper, will help the field significantly. I have some suggestions that I hope will be useful to improve the manuscript and the cell death community.

1. Can the authors also include the concentration in which the antibodies were used for both immunoblot and IHC? We apologise if this was not clear in our initial submission. The dilution of

primary antibodies used for immunoblot is stated on line 606: “1:2000 dilution for rat primary antibodies or 1:1000 for other primary antibodies”. The dilution of the primary antibodies used for immunohistochemistry is detailed in File **EV1**. The stock concentration of the antibodies is listed in the *Materials* section of the manuscript (except for the concentration of Cell Signaling Technology antibodies, which is often not disclosed).

In my experience, some antibodies such as the CST cleaved caspase-8 works much better when blocked with BSA instead of milk; is this also the case for these antibodies? If so, the authors should include the optimal blocking reagent for each antibody in the methods. We thank the reviewer for raising this important point. The only antibody used in this study that worked much better when diluted in BSA instead of milk was the Caspase-8 antibody (clone B.925.8). All other antibodies performed similarly for immunoblot when diluted in either milk or BSA. These details have now been added to line 604.

2. I am curious (and frankly a little sceptical) about the multiple necrosome puncta observed in HT-29 cells (Fig 5C) since inflammatory foci or innate immune signalling hubs typically collapse into one single puncta. This is observed for example during RIG-I/MAVS, TLR4 and inflammasome-ASC. Are the authors confident that these are not artefacts? We thank the reviewer for prompting further discussion on this important point. We and many others have shown that necroptotic signaling results in the formation of multiple distinct necrosomes per cell. Some published examples of this include Chen X. et al., 2022 *Nat. Cell Biol*, Samson A. et al., 2020 *Nat. Comms.*, Samson A. et al., 2021 *Cell Death Diff.* and Sun L. et al., 2012 *Cell*. We have referenced these papers on line 293.

3. I appreciate that patient samples are previous and limited. However, I find it hard to draw meaningful conclusion for Figure 6 the authors are basically comparing one healthy control to one IBD patient per group (please correct me if I am wrong). This will be much more convincing with a few more samples. We agree with the reviewer (and Reviewer 1’s similar comments) and have now added immunoblot data from two more patients (Fig. **EV8**) along with immunohistochemistry data for cleaved Caspase-3, Caspase-8, RIPK1, RIPK3 and MLKL from 6 patients (Fig. **EV9-10**). We believe this number of patients is sufficient to show that automated immunohistochemistry for the necroptotic pathway is feasible and offers spatial insight into necroptotic signaling. Despite providing this proof-of-concept, we agree with the reviewer that more work needs to be done to formally show that necrosome detection has clinical utility. To highlight this point, we have added the following to line 381: “future studies with a larger number of biopsies are needed to determine whether the immunoblot detection of necroptotic signaling significantly correlates with the immunohistochemical detection of Caspase-8+ clusters”.

4. In Fig 6D, using IHC, the authors found caspase-8 puncta in inflamed IBD patients and propose this as a potential unbiased method to detect the necrosome. I disagree with this definition since caspase-8 puncta can also represent the ripoptosome/Complex II that drives apoptosis. Yes, it is well-established that Caspase-8 oligomerisation occurs during both extrinsic apoptosis and necroptosis. Despite this understanding, our immunohistochemistry detected pronounced clusters of Caspase-8 in mouse dermal fibroblasts and HT29 cells undergoing necroptosis, but not in these same cell types undergoing apoptosis (Fig. **5** and **EV6**). Similarly, pronounced clusters of epithelial Caspase-8 were observed in biopsies from necroptosis-prone IBD patients, but not in biopsies from apoptosis-prone IBD patients where cleaved Caspase-3⁺ crypts were readily detected (Fig. **EV9-10**). Collectively, these data suggest that the immunohistochemical detection of Caspase-8⁺ clusters is indicative of necroptotic, rather than apoptotic signaling. Why the immunohistochemical detection of Caspase-8 distinguishes between necroptosis and apoptosis is unclear at this point, but this likely relates to the observation that the Caspase-8-containing complexes that form during necroptosis are qualitatively different to those that form during apoptosis. We have added text to line 375 to provide further explanation for our rationale in light of the reviewer’s comments.

Since the human MLKL antibodies seem to work in HT-29 cells (Fig. 5A), and yet only caspase-8 but MLKL puncta are not detected in human samples, I would instead conclude that inflamed IBD tissues are associated with apoptosis, not necroptosis (at least based on 1 patient sample). We do not know why clusters of Caspase-8, but not clusters of RIPK1 or MLKL, were detectable in biopsies from two patients with active necroptotic signaling (Fig. 6 and Fig. EV9-10). Our inability to detect clusters of MLKL in patients using immunohistochemistry does not suggest that apoptosis is the prevalent form of cell death. Instead, we believe this relates to a technical limitation (e.g. resolution limit) or to a gap in our understanding of how necroptosis manifests in vivo as mentioned on line 496.

Referee #3 (Remarks for Author):

This manuscript by Chiou et al describes a standardized process for immunohistochemical assessment of necroptosis mediators in murine and human tissue sections. The clinical implications of reproducibly characterizing cell death activity is an important goal. The authors optimize a number of antibodies and validate these antibodies. They use these reagents to examine the expression of several core cell death mediators, including caspase 8, RIP1, and RIP3. They localize differential expression of some of these markers to distinct cell subtypes in murine intestinal tissues, and note selective changes in response to TNF, antibiotics and to immunogens.

The general idea of optimizing immunohistochemical quantification of these important proteins is well reasoned, given the importance of these proteins to cellular survival. It is a little disappointing that MLKL could not be studied alongside RIP3, given the intriguing paradigm of RIP3 dependent MLKL necroptosis processes versus non-necroptosis RIP3 dependent processes. Standardization of these assays for use upon formalin fixed tissues could significantly enhance our understanding of the potential roles of necroptosis in various diseases. We agree wholeheartedly with the reviewer; we have spent the past 5 years seeking specific IHC antibodies for human RIPK3, including raising many of our own. Fortunately, during the review process, we were able to develop an automated immunohistochemical stain to specifically detect human RIPK3 in formalin-fixed cells and tissues using a new human RIPK3 antibody that has entered the market (new data in Fig. EV5 and EV7). We have now also compared the expression pattern of RIPK3 and MLKL in the human intestine (Fig. EV10). In addition, the data in Fig. EV4 directly compares the expression of RIPK3 and MLKL in the mouse spleen in naïve and immunised mice.

Issues that should be addressed include:

1. Expression levels of RIP3 were originally identified as conferring susceptibility to necroptosis by Xiaodang Wang's lab (He et al, Cell 2009). Chiou et al found selective expression of RIP3 in subsets of intestinal epithelial cells and splenocytes in unperturbed mice, reinforcing the idea that this protein may be particularly important in regulating basal homeostasis of these cells. We agree that this is an important point worth highlighting. As such, we have added the following to line 485: "This line of enquiry circles back to the idea that variations in RIPK3 expression are a key determinant of necroptotic potential (Cook et al., 2014; He et al., 2009; Najafov et al, 2018)."

As the authors associate increased RIP3 expression with robustly proliferating cells, and as necroptosis is experimentally increased when apoptosis mediators are inhibited (e.g., ZVAD blockade of caspases). Do the authors think that increased proliferation and caspase inhibition enhance necroptosis via similar/overlapping or distinct mechanisms. In the latter scenario, additive or synergistic necroptosis might be observed in certain scenarios. This issue might be inferred from histochemical data, but could also be tested experimentally. We appreciate that our manuscript provides in vivo support for a link between necroptotic potential and the cell cycle. However, the main purpose of our paper was to provide robust immunohistochemical methods to pinpoint necroptotic pathway expression in formalin-fixed mouse and human tissues. We feel that

experiments to explore the influence of cell cycling on necroptosis are beyond the scope of this study. To encourage future investigation into this topic, we have added the following to line 466: “It is noteworthy that certain non-mitotic cells, such as bone marrow-derived macrophages, are highly sensitive to necroptotic stimuli (Zelic & Kelliher, 2018). Thus, necroptosis is not reliant upon mitosis. Nonetheless, given the striking correlation between RIPK3 and mitotic markers, and since RIPK3’s activity and interactome vary considerably during the cell cycle (Gupta & Liu, 2021; Liccardi et al., 2019), future studies should investigate how cell proliferation influences necroptotic susceptibility.”

2. The complexity and activity of cell death pathways might be enhanced by broadening the panel to include other well characterized regulators of this process, e.g., phosphorylation of RIP3 is thought to reflect RIP3 activation during necroptosis signaling, and antibodies specific for phosphorylated RIP3 have been described. Could the authors test these antibodies to potentially enhance the detection of active necroptosis? We thank the reviewer for these ideas, which we have mulled over extensively ourselves. We have previously tried to use immunofluorescence to detect pRIPK3 (Samson et al., 2021 *CDD*). For mouse models, Genentech’s pRIPK3 antibody (clone GEN125-35-9) is a high-quality option with an accompanying paper outlining its use for immunohistochemistry (Webster et al., 2018 *Methods Mol Biol*). However, clone GEN125-35-9 is only available via a Materials Transfer Agreement which limits the quantities available for IHC applications. Unfortunately, we are yet to find a suitable antibody for imaging human pRIPK3. Abcam’s RIPK3 antibody (clone EPR9627) and Cell Signaling Technologies’ RIPK3 antibody (clone D6W2T) work for immunoblotting, but they cross-react strongly with other cellular proteins which precludes their use for imaging studies (Samson et al., 2021 *CDD*). Another salient point is that RIPK3 is autophosphorylated at S227 under basal conditions, with the degree of phosphorylation at S227 increasing under necroptotic conditions (Meng Y. et al., 2022 *CDDis*). Thus, mere presence of pRIPK3 is not indicative of necroptotic signaling, but instead relies on an increase in the levels of pS227 that is difficult to benchmark in biopsies where, unlike in cultured cells, no basal controls exist. Because of these considerations (and more reasons discussed in the paragraph starting on line 426), we have focused on detecting the non-phosphorylated form of RIPK3 via immunohistochemistry – for which there are no well-established protocols.

3. Along similar lines as point 2 above, it would be interesting to know whether phosphorylation of RIPK1 is increased. We understand that caspase 8 cleavage would be difficult to localize by immunoblotting, but there are antibodies specific for caspase 8 cleavage products. These are excellent queries, which we have sought to address ourselves unsuccessfully previously. We and others have shown that antibodies can be used to detect pRIPK1 (e.g. Samson et al., 2021 *CDD*). We have now developed an automated stain using a Cell Signaling Technology antibody (clone D813A) that specifically detects human pRIPK1 (**Response Figure 2**). Unfortunately, we have experienced major fluctuations in signal strength between different batches of clone D813A. For this reason, we have decided not to include this pRIPK1 protocol in this manuscript. Similarly, we were not able to establish an automated protocol for cleaved Caspase-8, but will continue trying to develop one for future studies.

Optimisation of pRIPK1 IHC on human intestinal biopsy

Response Figure 2. Representative micrographs of an IBD patient biopsy stained for pRIPK1 (clone D813A; left column) or non-phospho-RIPK1 (clone D94C12; right column). Indicative of specificity, the inset shows a pRIPK1 signal (top row) that is precluded by pre-treatment of the section with λ -phosphatase (bottom row). Note, the non-phospho-RIPK1 signal is unaltered by pre-treatment of the section with λ -phosphatase.

4. *The notion that RIP3 expression is an acute phase reactant is interesting, and might align with the cell biological notion of inflammatory cell death. Acute phase reactants have clinically been considered in the context of inflammatory serum markers (i.e., secreted proteins). Do the authors think that RIP3 might be detectable in serum, perhaps released from dying cells that are not quietly phagocytosed by neighboring cells?* We thank the reviewer for raising these important points. To address this question, during revision, we developed a sandwich ELISA using two monoclonal antibodies – clone 8G7 and clone 1H12 (both of which were previously reported and are available from MerckMillipore) – that recognise distinct epitopes in mouse RIPK3. Using this ELISA, we find that RIPK3 levels indeed increase in serum after TNF administration (Fig. 3K-L), but not when TNF and the necroptosis pathway inhibitor, Nec1s, were co-administered. As mentioned on line 211, these data suggest that TNF-induced RIPK1-dependent signaling leads to the release of RIPK3 into circulation. Whether this TNF-induced release of RIPK3 into circulation is due to necroptotic death is unknown, but seems likely.

5. *Systemic TNF injection induced RIPK3 expression in IECs, selected vascular beds and liver, while caspase 8 and RIPK1 expression were not (Fig. 3). This observation raises questions about the regulation of these proteins. Can the authors gain some insights into the mRNA expression levels of these genes via some type of spatial transcriptomic assay?* We do not know why the levels of RIPK3 are so dynamically regulated whereas the expression of Caspase-8, for example, is relatively constant. We agree that this is a very interesting question. However, addressing this question will likely involve detailed analyses of RIPK3's transcriptional and post-transcriptional regulation; experiments which we feel are beyond the scope of this manuscript.

6. *The shift of RIPK1 and RIPK3 expression within IEC subsets is interesting, and begs the question of whether antibiotic depletion alters IEC transit time. Could this be addressed via BrdU pulse expt?* Again, we thank the reviewer for prompting further discussion on an important point. It

is known that depletion of the microbiota, either by antibiotic-treatment or germ-free conditions, reduces epithelial turnover (Park J. et al., 2016 *PLoS ONE*). The possibility that reduced epithelial turnover influences RIPK1/3-clustering is now mentioned on line 239.

7. Heterogeneity in cell death signaling proteins in human IBD patients are very interesting. Can the authors provide some clinical information regarding the patients? Did the patients with increased signs of necroptosis display greater signs of intestinal or systemic inflammation? We apologise if this was not clear in our original submission and thank the reviewer for prompting clarification. The clinical information for each patient (including age, sex, disease subtype, biopsy location and treatment history) is provided in Table 1. In addition, Fig. **6B** shows the degree of cellular disease activity detected at each biopsy site as determined by blinded histopathological scoring. To assist readers, we have now added text to line 332 to ensure the draw further attention to this clinical information.

16th Apr 2024

Dear Dr. Samson,

Thank you for the submission of your revised manuscript to EMBO Molecular Medicine. I am pleased to inform you that we will be able to accept your manuscript pending the following final amendments:

- 1) Authors: We note a discrepancy of author's name: Christopher R. Horne in the ms file vs. Chris Horne in our submission system. Please correct.
- 2) Author checklist: Please submit a complete checklist. Currently response in the cell D81 is missing.
- 3) Figures: Up to 5 EV figures should be uploaded as individual figure files, with their legends added to the manuscript text after the main figure legends. The remaining figures should be renamed to "Appendix Figure S1" etc. and compiled with their legends in a PDF labelled Appendix. The appendix file has a table of content with page numbers on the title page. For more information on figure presentation please check "Author Guidelines".
<https://www.embopress.org/page/journal/17574684/authorguide#datapresentationformat>
- 4) In the main manuscript file, please do the following:
 - Please address all comments suggested by our data editors listed below:
 - o Figure legends:
 1. Please note that the figure legend 1a (iii) is incorrectly labelled as 1a (iv). This needs to be rectified.
 2. Please note that the combined legend for figures EV 4a-j is mislabeled as EV 4a-h. Also, the individual legend for figure EV 4i is mislabeled as EV 4h. This needs to be rectified.
 3. Please note that the error bars are not defined in the legends of figures EV 4c-j.
 4. Please note that for heatmap present in figures 1b-c; a numbered scale bar is not provided. This needs to be rectified.
 5. Please note that the scale bar needs to be defined for figures 3c, e, g, i.
 6. Please note that the black arrowheads are not defined in the legend of figure 5a. This needs to be rectified.
 - The manuscript sections should be in the following order: Title page - Abstract & Keywords - Introduction - Results - Discussion - Methods - Data Availability - Acknowledgments - Disclosure Statement & Competing Interests - References - Figure Legends - Tables with legends - Expanded View Figure Legends.
 - Add callout for Figure 5A.
 - Remove Abbreviations, Highlights and List of Expanded View (EV) Figures and Tables.
 - In M&M, provide the antibody dilutions that were used for each antibody.
 - Please include structured Methods section that includes a Reagents and Tools Table followed by a Methods and Protocols section. File EV1 seems to be a detailed protocol in table format, please add it to the "Appendix" and rename tables to "Appendix Table S1" etc. and update the callouts in the text. Please check "Author Guidelines" for more information and to download table templates. <https://www.embopress.org/page/journal/17574684/authorguide#structuredmethods>
 - Rename "Conflict of Interest" to "Disclosure Statement & Competing Interests". We updated our journal's competing interests policy in January 2022 and request authors to consider both actual and perceived competing interests. Please review the policy <https://www.embopress.org/competing-interests> and update your competing interests if necessary.
 - In M&M, provide the statement that in addition to the WMA Declaration of Helsinki the experiments also conformed to the principles set out in the Department of Health and Human Services Belmont Report.
 - In M&M, a statistical paragraph should reflect all information that you have filled in the Authors Checklist, especially regarding randomization, blinding, replication.
 - In data availability please remove the following "Customised image analysis macros used in this study are available upon request to the corresponding authors. The automated immunohistochemical staining protocols used in this study are available as File EV1." Use the following format to report the accession number of your RNA Seq data:

[data type]: [full name of the resource] [accession number/identifier] ([doi or URL or identifiers.org/DATABASE:ACCESSION])

Please check "Author Guidelines" for more information.

<https://www.embopress.org/page/journal/17574684/authorguide#availabilityofpublishedmaterial>

5) Table: Please add Table 1 to the main manuscript file.

6) Funding: Please fuse it with "Acknowledgement" and make sure that information about all sources of funding are complete in both our submission system and in the manuscript. Currently, the Independent Research Institutes Infrastructure Support Scheme 9000719), Victorian State Government Operational Infrastructure Support scheme, WEHI Handman PhD scholarship, the Avant Foundation, Crohn's and Colitis Australia and the University of Melbourne Scholarships, Anaxis Pharma Pty Ltd are missing in our submission system.

7) The Paper Explained: Please provide "The Paper Explained" and add it to the main manuscript text. Please check "Author Guidelines" for more information. <https://www.embopress.org/page/journal/17574684/authorguide#researcharticleguide>

8) Synopsis: Every published paper now includes a 'Synopsis' to further enhance discoverability. Synopses are displayed on the journal webpage and are freely accessible to all readers. They include separate synopsis image and synopsis text.

- Synopsis image: Please provide a striking image or visual abstract as a high-resolution jpeg file 550 px-wide x (250-400)-px

high to illustrate your article.

- Synopsis text: Please provide a short standfirst (maximum of 300 characters, including space) as well as 2-5 one sentence bullet points that summarise the paper as a .doc file. Please write the bullet points to summarise the key NEW findings. They should be designed to be complementary to the abstract - i.e. not repeat the same text. We encourage inclusion of key acronyms and quantitative information (maximum of 30 words / bullet point). Please use the passive voice.

9) Source Data: Please upload all requested source data files as one file per figure and complete the source data check list (attached).

10) For more information: This space should be used to list relevant web links for further consultation by our readers. Could you identify some relevant ones and provide such information as well? Some examples are patient associations, relevant databases, OMIM/proteins/genes links, author's websites, etc...

11) As part of the EMBO Publications transparent editorial process initiative (see our Editorial at <http://embomolmed.embopress.org/content/2/9/329>), EMBO Molecular Medicine will publish online a Review Process File (RPF) to accompany accepted manuscripts. This file will be published in conjunction with your paper and will include the anonymous referee reports, your point-by-point response and all pertinent correspondence relating to the manuscript. Let us know whether you agree with the publication of the RPF and as here, if you want to remove or not any figures from it prior to publication. Please note that the Authors checklist will be published at the end of the RPF.

12) Please provide a point-by-point letter INCLUDING my comments as well as the reviewer's reports and your detailed responses (as Word file).

I look forward to reading a new revised version of your manuscript as soon as possible.

Yours sincerely,

Zeljko Durdevic

*** Instructions to submit your revised manuscript ***

1) a .docx formatted version of the manuscript text (including Figure legends and tables)

2) Separate figure files*

3) supplemental information as Expanded View and/or Appendix. Please carefully check the authors guidelines for formatting Expanded view and Appendix figures and tables at <https://www.embopress.org/page/journal/17574684/authorguide#expandedview>

4) a letter INCLUDING the reviewer's reports and your detailed responses to their comments (as Word file).

5) The paper explained: EMBO Molecular Medicine articles are accompanied by a summary of the articles to emphasize the major findings in the paper and their medical implications for the non-specialist reader. Please provide a draft summary of your article highlighting

This may be edited to ensure that readers understand the significance and context of the research.

Please refer to any of our published articles for an example.

6) For more information: There is space at the end of each article to list relevant web links for further consultation by our readers. Could you identify some relevant ones and provide such information as well? Some examples are patient associations, relevant databases, OMIM/proteins/genes links, author's websites, etc...

7) Author contributions: the contribution of every author must be detailed in a separate section.

8) EMBO Molecular Medicine now requires a complete author checklist

(<https://www.embopress.org/page/journal/17574684/authorguide>) to be submitted with all revised manuscripts. Please use the checklist as guideline for the sort of information we need WITHIN the manuscript. The checklist should only be filled with page numbers where the information can be found. This is particularly important for animal reporting, antibody dilutions (missing) and exact values and n that should be indicated instead of a range.

9) Every published paper now includes a 'Synopsis' to further enhance discoverability. Synopses are displayed on the journal webpage and are freely accessible to all readers. They include a short stand first (maximum of 300 characters, including space) as well as 2-5 one sentence bullet points that summarise the paper. Please write the bullet points to summarise the key NEW findings. They should be designed to be complementary to the abstract - i.e. not repeat the same text. We encourage inclusion of key acronyms and quantitative information (maximum of 30 words / bullet point). Please use the passive voice. Please attach these in a separate file or send them by email, we will incorporate them accordingly.

You are also welcome to suggest a striking image or visual abstract to illustrate your article. If you do please provide a jpeg file 550 px-wide x 300-800px high.

10) A Conflict of Interest statement should be provided in the main text

11) Please note that we now mandate that all corresponding authors list an ORCID digital identifier. This takes <90 seconds to complete. We encourage all authors to supply an ORCID identifier, which will be linked to their name for unambiguous name identification.

Currently, our records indicate that the ORCID for your account is 0000-0002-0637-2716.

Link Not Available

Photos 400-800 DPI

*Additional important information regarding figures and illustrations can be found at

<https://bit.ly/EMBOPressFigurePreparationGuideline>. See also figure legend preparation guidelines:

<https://www.embopress.org/page/journal/17574684/authorguide#figureformat>

***** Reviewer's comments *****

Referee #1 (Remarks for Author):

The authors have done a thorough job addressing all comments raised. No further revisions are needed.

Referee #3 (Remarks for Author):

The authors have addressed our queries

The authors addressed the remaining editorial issues.

22nd Apr 2024

Dear Dr. Samson,

We are pleased to inform you that your manuscript is accepted for publication and is now being sent to our publisher to be included in the next available issue of EMBO Molecular Medicine.
